# The Crucial Role of Normalization in Sharpness-Aware Minimization

**Yan Dai** [*]
IIIS, Tsinghua University
yan-dai20@mails.tsinghua.edu.cn

**Kwangjun Ahn** [*]
EECS, MIT
kjahn@mit.edu

**Suvrit Sra**
TU Munich / MIT
suvrit@mit.edu

## Abstract

Sharpness-Aware Minimization (SAM) is a recently proposed gradient-based optimizer (Foret et al., ICLR 2021) that greatly improves the prediction performance of deep neural networks. Consequently, there has been a surge of interest in explaining its empirical success. We focus, in particular, on understanding *the role played by normalization*, a key component of the SAM updates. We theoretically and empirically study the effect of normalization in SAM for both convex and non-convex functions, revealing two key roles played by normalization: i) it helps in stabilizing the algorithm; and ii) it enables the algorithm to drift along a continuum (manifold) of minima – a property identified by recent theoretical works that is the key to better performance. We further argue that these two properties of normalization make SAM robust against the choice of hyper-parameters, supporting the practicality of SAM. Our conclusions are backed by various experiments.

## 1 Introduction

We study the recently proposed gradient-based optimization algorithm *Sharpness-Aware Minimization (SAM)* (Foret et al., 2021) that has shown impressive performance in training deep neural networks to generalize well (Foret et al., 2021; Bahri et al., 2022; Mi et al., 2022; Zhong et al., 2022). SAM updates involve an ostensibly small but key modification to Gradient Descent (GD). Specifically, for a loss function $\mathcal{L}$ and each iteration $t \geq 0$, instead of updating the parameter $w_t$ as $w_{t+1} = w_t - \eta\nabla\mathcal{L}(w_t)$ (where $\eta$ is called the *learning rate*), SAM performs the following update:[1]

$$w_{t+1} = w_t - \eta\nabla\mathcal{L}\left(w_t + \rho\frac{\nabla\mathcal{L}(w_t)}{\|\nabla\mathcal{L}(w_t)\|}\right), \tag{1}$$

where $\rho$ is an additional hyper-parameter that we call the *perturbation radius*. Foret et al. (2021) motivate SAM as an algorithm minimizing the robust loss $\max_{\|\epsilon\| \leq \rho} \mathcal{L}(w + \epsilon)$, which is roughly the loss at $w$ (i.e., $\mathcal{L}(w)$) plus the "sharpness" of the loss landscape around $w$, hence its name.

The empirical success of SAM has driven a recent surge of interest in characterizing its dynamics and theoretical properties (Bartlett et al., 2022; Wen et al., 2023; Ahn et al., 2023d). However, a major component of SAM remains unexplained in prior work: the role and impact of the normalization factor $\frac{1}{\|\nabla\mathcal{L}(w_t)\|}$ used by SAM. In fact, quite a few recent works drop the normalization factor for simplicity when analyzing SAM (Andriushchenko and Flammarion, 2022; Behdin and Mazumder, 2023; Agarwala and Dauphin, 2023; Kim et al., 2023; Compagnoni et al., 2023). Instead of the SAM update (1), these works consider the following update that we call *Un-normalized SAM (USAM)*:

$$w_{t+1} = w_t - \eta\nabla\mathcal{L}(w_t + \rho\nabla\mathcal{L}(w_t)). \tag{2}$$

---

[*]The first two authors contribute equally. Work done while Yan Dai was visiting MIT.

[1]In principle, the normalization in Equation 1 may make SAM ill-defined. However, Wen et al. (2023, Appendix B) showed that except for countably many learning rates, SAM (with any $\rho$) is always well-defined for almost all initialization. Hence, throughout the paper, we assume that the SAM iterates are always well-defined.

Apart from experimental justifications in (Andriushchenko and Flammarion, 2022), the effect of this simplification has not yet been carefully investigated, although it is already widely adopted in the community. Thus, is it really the case that such normalization can be omitted "for simplification" when theoretically analyzing SAM? These observations raise our main question:

*What is the role of the normalization factor $\frac{1}{\|\nabla \mathcal{L}(w_t)\|}$ in the SAM update* (1)*?*

## 1.1 Motivating Experiments and Our Contributions

We present our main findings through two motivating experiments. For the setting, we choose the well-known over-parameterized matrix sensing problem (Li et al., 2018); see Appendix A for details.

1. **Normalization helps with stability.** We first pick a learning rate $\eta$ that allows GD to converge, and we gradually increase $\rho$ from 0.001 to 0.1. Considering the early stage of training shown in Figure 1. One finds that *SAM has very similar behavior to GD*, whereas *USAM diverges even with a small $\rho$* – it seems that normalization helps stabilize the algorithm.

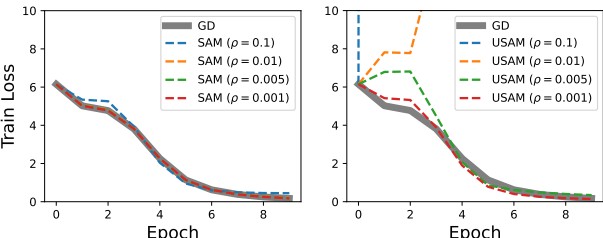

Figure 1: Role of normalization for stabilizing algorithms ($\eta = 0.05$).

2. **Normalization permits moving along minima.** We reduce the step size by 10 times and consider their performance of reducing test losses in the long run. One may regard Figure 2 as the behavior of SAM, USAM, and GD when close to a "manifold" of minima (which exists since the problem is over-parametrized) as the training losses are close to zero. The first plot compares SAM and USAM with the same $\rho = 0.1$ (the largest $\rho$ for which USAM doesn't diverge): notice that USAM and GD both converge to a minimum and do not move further; on the other hand, SAM keeps decreasing the test loss, showing its ability to drift along the manifold. We also vary $\rho$ and compare their behaviors (shown on the right): *the ability of SAM to travel along the manifold of minimizers seems to be robust* to the size of $\rho$, while *USAM easily gets stuck at a minimum*.

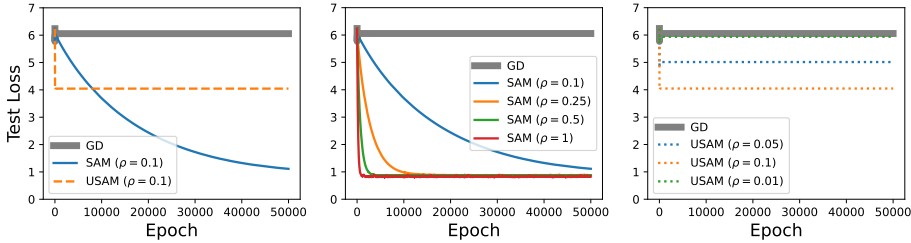

Figure 2: Role of normalization when close to a manifold of minimizers ($\eta = 0.005$).

**Overview of our contributions.** In this work, as motivated by Figure 1 and Figure 2, we identify and theoretically explain the two roles of normalization in SAM. The paper is organized as follows.

1. In Section 2, we study the role of normalization in the algorithm's stability and show that normalization helps stabilize. In particular, we demonstrate that normalization ensures that GD's convergence implies SAM's non-divergence, whereas USAM can start diverging much earlier.

2. In Section 3, we study the role of normalization near a manifold of minimizers and show that the normalization factor allows iterates to keep drifting along this manifold – giving better performance in many cases. Without normalization, the algorithm easily gets stuck and no longer progresses.

3. In Section 4, to illustrate our main findings, we adopt the sparse coding example of Ahn et al. (2023a). Their result implies a dilemma in hyper-parameter tuning for GD: a small $\eta$ gives worse performance, but a large $\eta$ results in divergence. We show that this dilemma extends to USAM – but not SAM. In other words, SAM easily solves the problem where GD and USAM often fail.

These findings also shed new light on why SAM is practical and successful, as we highlight below.

**Practical importance of our results.** The main findings in this work explain and underscore several practical aspects of SAM that are mainly due to the normalization step. One practical feature of SAM is the way the hyper-parameter $\rho$ is tuned: Foret et al. (2021) suggest that $\rho$ can be tuned independently after tuning the parameters of base optimizers (including learning rate $\eta$, momentum $\beta$, and so on). In particular, this feature makes SAM a perfect "add-on" to existing gradient-based optimizers. Our findings precisely support this practical aspect of SAM. Our results suggest that ***the stability of SAM is less sensitive to the choice of $\rho$, thanks to the normalization factor.***

The same principle applies to the behavior of the algorithm near the minima: Recent theoretical works (Bartlett et al., 2022; Wen et al., 2023; Ahn et al., 2023d) have shown that the drift along the manifold of minimizers is a main feature that enables SAM to reduce the sharpness of the solution (which is believed to boost generalization ability in practice) – our results indicate that ***the ability of SAM to keep drifting along the manifold is independent of the choice of $\rho$, again owing to normalization.*** Hence, our work suggests that the normalization factor plays an important role towards SAM's empirical success.

## 1.2 Related Work

**Sharpness-Aware Optimizers.** Inspired by the empirical and theoretical observation that the generalization effect of a deep neural network is correlated with the "sharpness" of the loss landscape (see (Keskar et al., 2017; Jastrzkebski et al., 2017; Jiang et al., 2020) for empirical observations and (Dziugaite and Roy, 2017; Neyshabur et al., 2017) for theoretical justifications), several recent papers (Foret et al., 2021; Zheng et al., 2021; Wu et al., 2020) propose optimizers that penalize the sharpness for the sake of better generalization. Subsequent efforts were made on making such optimizers scale-invariant (Kwon et al., 2021), more efficient (Liu et al., 2022; Du et al., 2022), and generalize better (Zhuang et al., 2022). This paper focuses on the vanilla version proposed by Foret et al. (2021).

**Theoretical Advances on SAM.** Despite the success of SAM in practice, theoretical understanding of SAM was absent until two recent works: Bartlett et al. (2022) analyze SAM on locally quadratic losses and identify a component reducing the sharpness $\lambda_{\max}(\nabla^2 \mathcal{L}(w_t))$, while Wen et al. (2023) characterize SAM near the manifold $\Gamma$ of minimizers and show that SAM follows a Riemannian gradient flow reducing $\lambda_{\max}(\nabla^2 \mathcal{L}(w))$ when i) initialized near $\Gamma$, and ii) $\eta$ is "small enough". Note that while the results of Wen et al. (2023) apply to more general loss functions, our result in Theorem 16 applies to i) any initialization far from the origin, and ii) any $\eta = o(1)$ and $\rho = O(1)$. A recent work by Ahn et al. (2023d) formulates the notion of $\varepsilon$-approximate flat minima and analyzed the iteration complexity of practical algorithms like SAM to find such approximate flat minima. A concurrent work by Si and Yun (2023) also analyzes the original version of SAM with the normalization in (1), and makes a case that practical SAM does not converge all the way to optima.

**Unnormalized SAM (USAM).** USAM was first proposed by Andriushchenko and Flammarion (2022) who observed a similar performance between USAM and SAM when training ResNet over CIFAR-10. This simplification is further accepted by Behdin and Mazumder (2023) who study the regularization effect of USAM over a linear regression model, by Agarwala and Dauphin (2023) who study the initial and final dynamics of USAM over a quadratic regression model, and by Kim et al. (2023) who study the convergence instability of USAM near saddle points. To our knowledge, (Compagnoni et al., 2023) is the only work comparing SAM and USAM dynamics. More preciously, they consider the continuous-time behavior of SGD, SAM, and USAM and find different behaviors of SAM and USAM: USAM attracts local minima while SAM aims at global ones. Still, we remark that as they are considering continuous-time variants of algorithms while we consider discrete (original) versions, our results directly apply to the SAM deployed in practice and the USAM studied in theory.

**Edge-of-Stability.** In the optimization theory literature, Gradient Descent (GD) was only shown to find minima if the learning rate $\eta$ is smaller than an "Edge-of-Stability" threshold, which is related to the sharpness of the nearest minimum. However, people recently observe that when training neural networks, GD with a $\eta$ much larger than that threshold often finds good minima as well (see (Cohen

et al., 2021) and references therein). Aside from convergence, GD with large $\eta$ is also shown to find *flatter* minima (Arora et al., 2022; Ahn et al., 2022; Wang et al., 2022; Damian et al., 2023).

## 2 Role of Normalization for Stability

In this section, we discuss the role of normalization in the stability of the algorithm. We begin by recalling a well-known fact about the stability of GD: for a convex quadratic cost with the largest eigenvalue of Hessian being $\beta$ (i.e., $\beta$-smooth), GD converges to a minimum iff $\eta < {}^2/_\beta$. Given this folklore fact, we ask: how do the ascent steps in SAM (1) and USAM (2) affect their stability?

### 2.1 Strongly Convex and Smooth Losses

Consider an $\alpha$-strongly-convex and $\beta$-smooth loss function $\mathcal{L}$ where GD is guaranteed to converge once $\eta < {}^2/_\beta$. We characterize the stability of SAM and USAM in the following result.

**Theorem 1** (Strongly Convex and Smooth Losses). *For any $\alpha$-strongly-convex and $\beta$-smooth loss function $\mathcal{L}$, for any learning rate $\eta < {}^2/_\beta$ and perturbation radius $\rho \geq 0$, the following holds:*

*1. **SAM.** The iterate $w_t$ converges to a local neighborhood around the minimizer $w^\star$. Formally,*

$$\mathcal{L}(w_t) - \mathcal{L}(w^\star) \leq \big(1 - \alpha\eta(2 - \eta\beta)\big)^t(\mathcal{L}(w_0) - \mathcal{L}(w^\star)) + \frac{\eta\beta^3\rho^2}{2\alpha(2 - \eta\beta)}, \quad \forall t. \tag{3}$$

*2. **USAM.** In contrast, there exists some $\alpha$-strongly-convex and $\beta$-smooth loss $\mathcal{L}$ such that the USAM with $\eta \in \big({}^2/_{(\beta+\rho\beta^2)}, {}^2/_\beta\big]$ diverges for all except measure zero initialization $w_0$.*

As we discussed, it is well-known that GD converges iff $\eta < {}^2/_\beta$, and Theorem 1 shows that SAM also does not diverge and stays within an $\mathcal{O}(\sqrt{\eta}\rho)$-neighborhood around the minimum as long as $\eta < {}^2/_\beta$. However, USAM diverges with an even lower learning rate: $\eta > {}^2/_{(\beta+\rho\beta^2)}$ can already make USAM diverge. Intuitively, the larger the value of $\rho$, the easier it is for USAM to diverge.

One may notice that Equation 3, compared to the standard convergence rate of GD, exhibits an additive bias term of order $\mathcal{O}(\eta\rho^2)$. This term arises from the unstable nature of SAM: the perturbation in (1) (which always has norm $\rho$) prevents SAM from decreasing the loss monotonically. Thus, SAM can only approach a minimum up to a neighborhood. For this reason, in this paper whenever we say SAM "finds" a minimum, we mean its iterates approach and stay within a neighborhood of that minimum.

Due to space limitations, the full proof is postponed to Appendix C and we only outline it here.

*Proof Sketch.* For SAM, we show an analog to the descent lemma of GD as follows (see Lemma 9):

$$\mathcal{L}(w_{t+1}) \leq \mathcal{L}(w_t) - \frac{1}{2}\eta(2 - \eta\beta)\|\nabla\mathcal{L}(w_t)\|^2 + \frac{\eta^2\beta^3\rho^2}{2}. \tag{4}$$

By invoking the strong convexity that gives $\mathcal{L}(w_t) - \mathcal{L}(w^*) \leq \frac{1}{2\alpha}\|\nabla f(w_t)\|^2$, we obtain

$$\mathcal{L}(w_{t+1}) - \mathcal{L}(w^\star) \leq \big(1 - \alpha\eta(2 - \eta\beta)\big)(\mathcal{L}(w_t) - \mathcal{L}(w^\star)) + \frac{\eta^2\beta^3\rho^2}{2}.$$

Recursively applying this relation gives the first conclusion. For USAM, we consider the quadratic loss function same as (Bartlett et al., 2022). Formally, suppose that $\mathcal{L}(w) = \frac{1}{2}w^\top\Lambda w$ where $\Lambda = \mathrm{diag}(\lambda_1, \lambda_2, \ldots, \lambda_d)$ is a PSD matrix such that $\lambda_1 > \lambda_2 \geq \cdots \lambda_d > 0$. Let the eigenvectors corresponding to $\lambda_1, \lambda_2, \ldots, \lambda_d$ be $e_1, e_2, \ldots, e_d$, respectively. Then we show the following in Theorem 10: for any $\eta(\lambda_1 + \rho\lambda_1^2) > 2$ and $\langle w_0, e_1 \rangle \neq 0$, USAM must diverge. As $\mathcal{L}(w) = \frac{1}{2}w^\top\Lambda w$ is $\lambda_1$-smooth and $\lambda_d$-strongly-convex, the second conclusion also follows. $\square$

Intuitively, the difference in stability can be interpreted as follows: during the early stage of training, $w_t$ and $\nabla\mathcal{L}(w_t)$ often have large norms. The normalization in SAM then makes the ascent step $w_t + \rho\frac{\nabla\mathcal{L}(w_t)}{\|\nabla\mathcal{L}(w_t)\|}$ not too far away from $w_t$. Hence, if GD does not diverge for this $\eta$, SAM does not either (unless the $\rho$-perturbation is non-negligible, i.e., $\|w_t\| \gg \rho$ no longer holds). This is not true for USAM: since the ascent step is un-normalized, it leads to a point far away from $w_t$, making the size of USAM updates much larger. In other words, the removal of normalization leads to much more aggressive steps, resulting in a different behavior than GD and also an easier divergence.

## 2.2 Generalizing to Non-Convex Cases: Scalar Factorization Problem

Now let us move on to non-convex losses. We consider a *scalar version* of the matrix factorization problem $\min_{U,V} \frac{1}{2}\|UV^T - A\|_2^2$, whose loss function is defined as $\mathcal{L}(x,y) = \frac{1}{2}(xy)^2$. Denote the initialization by $(x_0, y_0)$, then $\mathcal{L}(x,y)$ is $\beta \triangleq (x_0^2 + y_0^2)$-smooth inside the region $\{(x,y) : x^2 + y^2 \leq \beta\}$. Hence, a learning rate $\eta < 2/\beta$ again allows GD to converge due to the well-known descent lemma. The following result compares the behavior of SAM and USAM under this setup.

**Theorem 2** (Scalar Factorization Problem; Informal). *For the loss function $\mathcal{L}(x,y) = \frac{1}{2}(xy)^2$ restricted to a $\beta$-smooth region, if we set $\eta = 1/\beta < 2/\beta$ (so GD finds a minimum), then*

1. **SAM.** *SAM never diverges and approaches a minimum within an $O(\rho)$-neighborhood (in fact, SAM with distinct $\rho$'s always find the same minimum $(0,0)$).*

2. **USAM.** *On the other hand, USAM diverges once $\rho \geq 15\eta$ – which could be much smaller than 1.*

Thus, our observation in Theorem 1 is not limited to convex losses – for our non-convex scalar-factorization problem, the stability of SAM remains robust to the choice of $\rho$, while USAM is provably unstable. One may refer to Appendix D for the formal statement and proof of Theorem 2.

## 2.3 Experiment: Early-Stage Behaviors when Training Neural Networks

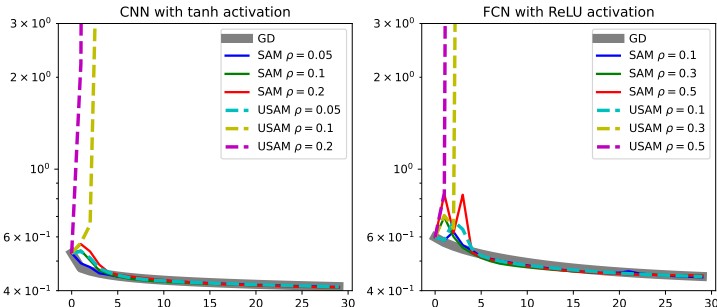

Figure 3: Behaviors of different algorithms when training neural networks ($\eta = 0.025$).

As advertised, our result holds not only for convex or toy loss functions but also for practical neural networks. In Figure 3, we plot the early-stage behavior of GD, SAM, and USAM with different $\rho$ values (while fixing $\eta$). We pick two neural networks: a convolutional neural network (CNN) with tanh activation and a fully-connected network (FCN) with ReLU activation. We train them over the CIFAR10 dataset and report the early-stage training losses. Similar to Figure 1, Theorem 1 and Theorem 2, *the stability of SAM is not sensitive to the choice of $\rho$, while USAM diverges easily*.

## 3 Role of Normalization for Drifting Near Minima

Now, we explain the second role of normalization: enabling the algorithm to drift near minima. To convince why this is beneficial, we adopt a loss function recently considered by Ahn et al. (2023a) when understanding the behavior of GD with large learning rates. Their result suggests that GD needs a "large enough" $\eta$ for enhanced performance, but this threshold can never be known a-priori in practice. To verify our observations from Figure 2, we study the dynamics of SAM and USAM over the same loss function and find that: i) *no careful tuning is needed for SAM*; instead, SAM with any configuration finds the same minimum (which is the "best" one according to Ahn et al. (2023a)); and ii) *such property is only enjoyed by SAM* – for USAM, careful tuning remains essential.

### 3.1 Toy Model: Single-Neuron Linear Network Model

To theoretically study the role of normalization near minima, we consider the simple two-dimensional non-convex loss $\mathcal{L}(x,y)$ defined over all $(x,y) \in \mathbb{R}^2$ as

$$\mathcal{L} : (x,y) \mapsto \ell(x \times y), \qquad \text{where } \ell \text{ is convex, even, and 1-Lipschitz.} \tag{5}$$

This $\mathcal{L}$ was recently studied in (Ahn et al., 2023a) to understand the behavior of GD with large $\eta$'s. By direct calculation, the gradient and Hessian of $\mathcal{L}$ at a given $(x, y)$ can be written as:

$$\nabla \mathcal{L}(x, y) = \ell'(xy) \begin{bmatrix} y \\ x \end{bmatrix}, \quad \nabla^2 \mathcal{L}(x, y) = \ell''(xy) \begin{bmatrix} y \\ x \end{bmatrix}^{\otimes 2} + \ell'(xy) \begin{bmatrix} 0 & 1 \\ 1 & 0 \end{bmatrix}. \tag{6}$$

Without loss of generality, one may assume $\ell$ is minimized at 0 (see Appendix E for more details regarding $\ell$). Then, $\mathcal{L}$ achieves minimum at the entire $x$- and $y$-axes, making it a good toy model for studying the behavior of algorithms near a continuum of minima. Finally, note that the parametrization $x \times y$ can be interpreted as a single-neuron linear network model – hence its name.

Before moving on to SAM and USAM we first briefly introduce the behavior of GD on such loss functions characterized in (Ahn et al., 2023a). Since $\ell$ is even, without loss of generality, we always assume that the initialization $w_0 = (x_0, y_0)$ satisfies $y_0 \geq x_0 > 0$.

**Theorem 3** (Theorems 5 and 6 of (Ahn et al., 2023a); Informal). *For any $\eta = \gamma/(y_0^2 - x_0^2)$, the GD trajectory over the loss function $\mathcal{L}(x, y) = \ell(xy)$ has two possible limiting points:*

*1. If $\gamma < 2$, then the iterates converge to $(0, y_\infty)$ where $y_\infty^2 \in [\gamma/\eta - \mathcal{O}(\gamma) - \mathcal{O}(\eta/\gamma), \gamma/\eta + \mathcal{O}(\eta/\gamma)]$.*

*2. If $\gamma > 2$, then the iterates converge to $(0, y_\infty)$ where $y_\infty^2 \in [2/\eta - \mathcal{O}(\eta), 2/\eta]$.*

Intuitively, the limiting point of GD (denoted by $(0, y_\infty)$) satisfies $y_\infty^2 \approx \min\{y_0^2 - x_0^2, 2/\eta\}$. For simplicity, we denote $\eta_{GD} \approx 2/y_0^2 - x_0^2$ as the threshold of $\eta$ that distinguishes these two cases.

**Interpretation of Ahn et al. (2023a).** Fixing the initialization $(x_0, y_0)$, it turns out this model has a nice connection to the sparse coding problem, wherein it's desirable to get a smaller $y_\infty^2$ (which we will briefly discuss in Section 4). According to Theorem 3, to get a smaller $y_\infty^2$, one must increase the learning rate $\eta$ beyond $\eta_{GD}$. Hence we mainly focus on the case where $\eta > \eta_{GD}$ – in which case we abbreviate $y_\infty^2 \approx 2/\eta$ (see Table 1). However, GD diverges once $\eta$ is too large – in their language, $\gamma$ cannot be much larger than 2. This dilemma of tuning $\eta$, as we shall illustrate in Section 4 in more detail, makes GD a brittle choice for obtaining a better $y_\infty^2$.

On the other hand, from the numerical illustrations in Figure 4, one can see that *SAM keeps moving along the manifold of minimizers* (i.e., the $y$-axis) until the origin. This phenomenon is characterized in Theorem 4; in short, any moderate choice of $\eta$ and $\rho$ suffices to drive SAM toward the origin – no difficult tuning needed anymore!

In contrast, USAM does not keep moving along the axis. Instead, a lower bound on $y_\infty^2$ also presents – although smaller than the GD version. As we will justify in Theorem 5, *USAM does get trapped* at some non-zero $y_\infty^2$. Thus, a dilemma similar to that of GD shows up: for enhanced performance, an aggressive $(\eta, \rho)$ is needed; however, as we saw from Section 2, this easily results in a divergence.

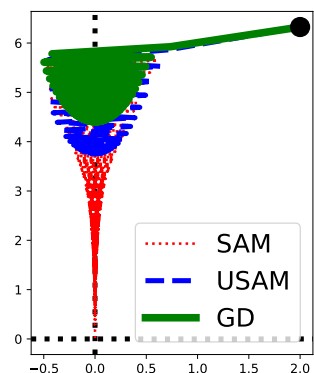

Figure 4: Trajectories of different algorithms for the $\ell(xy)$ loss ($\eta = 0.4$ and $\rho = 0.1$; initialization $(x_0, y_0) = (2, \sqrt{40})$ is marked by a black dot).

**Assumptions.** To directly compare with (Ahn et al., 2023a), we focus on the cases where $y_0^2 - x_0^2 = \gamma/\eta$ and $\gamma \in [\frac{1}{2}, 2]$ is a constant of moderate size; hence, $\eta$ is not too different from the $\eta_{GD}$ defined in Theorem 3. In contrast to most prior works which assume a tiny $\rho$ (e.g., (Wen et al., 2023)), we allow $\rho$ to be as large as a constant (i.e., we only require $\rho = \mathcal{O}(1)$ in Theorem 4 and Theorem 5).

### 3.1.1 SAM Keeps Drifting Toward the Origin

We characterize the trajectory of SAM when applied to the loss defined in Equation 5 as follows:

**Theorem 4** (SAM over Single-Neuron Networks; Informal). *For any $\eta \in [\frac{1}{2}\eta_{GD}, 2\eta_{GD}]$ and $\rho = \mathcal{O}(1)$, the SAM trajectory over the loss function $\mathcal{L}(x, y) = \ell(xy)$ can be divided into three phases:*

*1. **Initial Phase.** $x_t$ drops so rapidly that $|x_t| = \mathcal{O}(\sqrt{\eta})$ in $\mathcal{O}(1/\eta)$ steps. Meanwhile, $y_t$ remains large: specifically, $y_t = \Omega(\sqrt{1/\eta})$. Thus, SAM approaches the $y$-axis (the set of global minima).*

Table 1: Limiting points of GD, SAM, and USAM for the $\ell(xy)$ loss (assuming $\eta > \eta_{\text{GD}}$).

| Algorithm | GD | SAM | USAM |
|---|---|---|---|
| Limiting Point $(0, y_\infty)$ | $y_\infty^2 \approx 2/\eta$ | $y_\infty^2 \approx 0$ | $(1 + \rho y_\infty^2)y_\infty^2 \approx 2/\eta$ |

2. **Middle Phase.** $x_t$ *oscillates closely to the axis such that* $|x_t| = \mathcal{O}(\sqrt{\eta})$ *always holds. Meanwhile,* $y_t$ *decreases fast until* $y_t \leq |x_t|^2$ – *that is,* $|x_t|$ *remains small and SAM approaches the origin.*

3. **Final Phase.** $w_t = (x_t, y_t)$ *gets close to the origin such that* $|x_t|, |y_t| = \mathcal{O}(\sqrt{\eta} + \eta\rho)$. *We then show that* $w_t$ *remains in this region for the subsequent iterates.*

Informally, SAM first approaches the minimizers on $y$-axis (which form a manifold) and then keeps moving until a specific minimum. Moreover, SAM always approaches this minimum for almost all $(\eta, \rho)$'s. This matches our motivating experiment in Figure 2: No matter what hyper-parameters are chosen, SAM *always* drift along the set of minima, in contrast to the behavior of GD. This property allows SAM always to approach the origin $(0, 0)$ and remains in its neighborhood, while GD converges to $(0, \sqrt{2/\eta})$ (see Table 1). The formal version of Theorem 4 is in Appendix F.

### 3.1.2 USAM Gets Trapped at Different Minima

We move on to characterize the dynamics of USAM near the minima. Like GD or SAM, the first few iterations of USAM drive iterates to the $y$-axis. However, unlike SAM, USAM does not keep drifting along the $y$-axis and stops at some threshold – in the result below, we prove a lower bound on $y_t^2$ that depends on both $\eta$ and $\rho$. In other words, the lack of normalization factor leads to diminishing drift.

**Theorem 5** (USAM over Single-Neuron Networks; Informal)**.** *For any* $\eta \in [\frac{1}{2}\eta_{GD}, 2\eta_{GD}]$ *and* $\rho = \mathcal{O}(1)$, *the USAM trajectory over the loss function* $\mathcal{L}(x, y) = \ell(xy)$ *have the following properties:*

1. **Initial Phase.** *Similar to Initial Phase of* Theorem 4, $|x_t| = \mathcal{O}(\sqrt{\eta})$ *and* $y_t = \Omega(\sqrt{1/\eta})$ *hold for the first* $\mathcal{O}(1/\eta)$ *steps. That is, USAM also approaches* $y$-*axis, the set of global minima.*

2. **Final Phase.** *However, for USAM, once the following condition holds for some round* $\mathfrak{t}$:[3]

$$(1 + \rho y_\mathfrak{t}^2)y_\mathfrak{t}^2 < \frac{2}{\eta}, \quad i.e., \;\; y_\mathfrak{t}^2 < \tilde{y}_{USAM}^2 \triangleq \left(\sqrt{8\frac{\rho}{\eta} + 1} - 1\right)\Big/ 2\rho, \tag{7}$$

$|x_t|$ *decays exponentially fast, which in turn ensures* $y_\infty^2 \triangleq \liminf_{t\to\infty} y_t^2 \gtrsim (1 - \eta^2 - \rho^2)y_\mathfrak{t}^2$.

**Remark.** Note that USAM becomes GD as we send $\rho \to 0+$, and our characterized threshold $y_{\text{USAM}}^2$ indeed recovers that of GD (i.e., $2/\eta$ from Theorem 3) because $\lim_{\rho\to0+}(\sqrt{8\rho/\eta + 1} - 1)/2\rho = 2/\eta$.

Compared with SAM, the main difference occurs when close to minima, i.e., $|x_t| = \mathcal{O}(\sqrt{\eta})$. Consistent with our motivating experiment (Figure 2), the removal of normalization leads to diminishing drift along the minima. Thus, USAM is more like an improved version of GD rather than a simplification of SAM, and the comparison between Theorem 3 and Theorem 5 reveals that USAM only improves over GD if $\rho$ is large enough – in which case USAM is prone to diverges as we discussed in Section 2.

See Appendix G for a formal version of Theorem 5 together with its proof.

### 3.1.3 Technical Distinctions Between GD, SAM, and USAM

Before moving on, we present a more technical comparison between the results stated in Theorem 3 versus Theorem 4 and Theorem 5. We start with an intuitive explanation of why GD and USAM get stuck near the manifold of minima but SAM does not: when the iterates approach the set of minima, both $w_t$ and $\nabla\mathcal{L}(w_t)$ become small. Hence the normalization plays an important role: as $\nabla\mathcal{L}(w_t)$ are small, $w_t$ and $w_t + \rho\nabla\mathcal{L}(w_t)$ become nearly identical, which leads to a diminishing updates of GD and USAM near the minima. On the other hand, having the normalization term, the SAM update doesn't diminish, which prevents SAM from converging to a minimum and keeps drifting along the manifold.

---

[2]Specifically, either an exponential decay $y_{t+1} \lesssim (1 - \eta\rho^2)y_t$ or a constant drop $y_{t+1} \lesssim y_t - \eta\rho$ appears

[3]The $\tilde{y}_{\text{USAM}}^2$ in Equation 7 is defined as the solution to the equation $(1 + \rho y^2)y^2 = 2$.

This high-level idea is supported by the following calculation: recall Equation 6 that $\nabla\mathcal{L}(x_t, y_t) = \ell'(x_t y_t)\cdot[y_t \quad x_t]^\top$. Hence, when $|x_t| \ll y_t$ in Final Phase, the "ascent gradient" direction $\nabla\mathcal{L}(x_t, y_t)$ (i.e., the ascent steps in Equation 1 and Equation 2) is almost perpendicular to the $y$-axis. We thus rewrite the update direction (i.e., the difference between $w_{t+1}$ and $w_t$) for each algorithm as follows.

1. For SAM, after normalization, $\frac{\nabla\mathcal{L}(w_t)}{\|\nabla\mathcal{L}(w_t)\|}$ is roughly a unit vector along the $x$-axis. Hence, the update direction is the gradient at $w_{t+1/2} \approx [\rho \quad y_t]^\top$. Once $y_t$ is large (making $w_{t+1/2}$ far from minima), $\nabla\mathcal{L}(w_{t+1/2})$ thus have a large component along $y_t$, which leads to drifting near minima.

2. For GD, by approximating $\ell'(u) \approx u$, we derive $\nabla\mathcal{L}(x_t, y_t) \approx [x_t y_t^2 \quad x_t^2 y_t]^\top$. When $2/\eta > y_t^2$, the magnitude of $x_t$ is updated as $|x_{t+1}| \approx |x_t - \eta x_t y_t^2| = |(1 - \eta y_t^2)x_t|$, which allows an exponential decay. Thus, GD converges to a minimum and stop moving soon after $2/\eta > y_t^2$.

3. For USAM, the descent gradient is taken at $w_t + \rho\nabla\mathcal{L}(w_t) \approx [(1+\rho y_t^2)x_t \quad (1+\rho x_t^2)y_t]^\top$. Thus, $\nabla\mathcal{L}(w_t + \rho\nabla\mathcal{L}(w_t)) \approx [(1+\rho y_t^2)(1+\rho x_t^2)^2 x_t y_t^2 \quad (1+\rho y_t^2)^2(1+\rho x_t^2)x_t^2 y_t]^\top$ by writing $\ell'(u) \approx u$. This makes USAM deviate away from SAM and behave like GD: by the similar argument as GD, USAM stops at a minimum soon after $2/\eta > (1 + \rho y_t^2)(1 + \rho x_t^2)^2 y_t^2 \approx (1 + \rho y_t^2)y_t^2$!

Hence, the normalization factor in the ascent gradient helps maintain a non-diminishing component along the minima, leading SAM to keep drifting. This distinguishes SAM from GD and USAM.

## 3.2 USAM Gets Trapped Once Close to Minima

In this section, we extend our arguments to nonconvex costs satisfying Polyak-Lojasiewicz (PL) functions (see, e.g., (Karimi et al., 2016)). Recall that $f$ satisfies the $\mu$-PL condition if $\frac{1}{2}\|\nabla\mathcal{L}(w)\|^2 \geq \mu(\mathcal{L}(w) - \min_w \mathcal{L}(w))$ for all $w$. Building upon the analysis of Andriushchenko and Flammarion (2022), we show the following result when applying USAM to $\beta$-smooth and $\mu$-PL losses.

**Theorem 6** (USAM over PL Losses; Informal). *For $\beta$-smooth and $\mu$-PL loss $\mathcal{L}$, for any $\eta < 1/\beta$ and $\rho < 1/\beta$, and for any initialization $w_0$, $\|w_t - w_0\| \leq poly(\eta, \rho, \beta, \mu) \cdot \sqrt{\mathcal{L}(w_0) - \min_w \mathcal{L}(w)}$, $\forall t$.*

This theorem has the following consequence: Suppose that USAM encounters a point $w_0$ that is close to some minimum (i.e., $\mathcal{L}(w_0) \approx \min_w \mathcal{L}(w)$) during training. Then Theorem 6 implies that ***the total distance traveled by USAM from $w_0$ is bounded*** – in other words, the distance USAM moves along the manifold of minimizers can only be of order $\mathcal{O}(\sqrt{\mathcal{L}(w_0) - \min_w \mathcal{L}(w)})$.

As a remark, we compare Theorem 6 with the recent result by Wen et al. (2023): their result essentially implies that, for small enough $\eta$ and $\rho$, SAM iterates initialized close to a manifold of the minimizers approximately track some continuous dynamics (more precisely, a Riemannian gradient flow induced by a "sharpness" measure they find) and keep drifting along the manifold. This property is indeed in sharp contrast with USAM whose total travel distance is bounded.

The formal statement and proof of Theorem 6 are contained in Appendix H.

## 3.3 Experiments for Practical Neural Networking Training

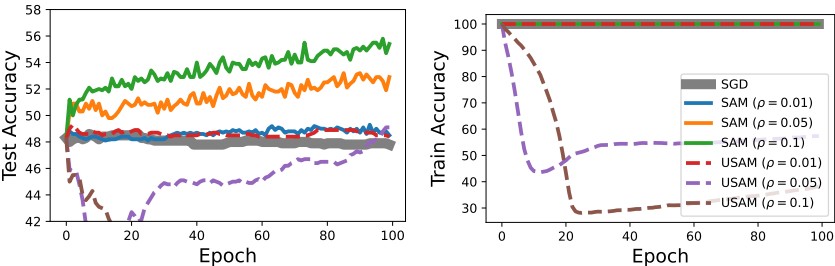

Figure 5: Training ResNet18 on CIFAR-10 from a bad global minimum ($\eta = 0.001$, batch size $= 128$).

We close this section by verifying our claims in practical neural network training. We train a ResNet18 on the CIFAR-10 dataset, initialized from a poor global minimum generated as per Liu et al. (2020)

(we used the "adversarial checkpoint" released by Damian et al. (2021)). This initialization has 100% training accuracy but only 48% test accuracy – which lets us observe a more pronounced algorithmic behavior near the minima via tracking the test accuracy. From Figure 5, we observe:

1. GD gets stuck at this adversarial minimum, in the sense that the test accuracy stays ats 48%.
2. SAM keeps drifting while staying close to the manifold of minimizers (because the training accuracy remains 100%), which results in better solutions (i.e., the test accuracy keeps increasing).
3. USAM with small $\rho$ gets stuck like GD, while USAM with larger $\rho$'s deviate from this manifold.

Hence, USAM faces the dilemma that we describe in Subsection 3.1: a conservative hyper-parameter configuration does not lead to much drift along the minima, while a more aggressive choice easily leads to divergence. However, the stability of SAM is quite robust to the choice of hyper-parameter and they all seem to lead to consistent drift along the minima.

**Remark.** Apart from the "adversarial checkpoint" which is unrealistic but can help highlight different algorithms' behavior when they are close to a bad minimum, we also conduct the same experiments but instead initialized from a "full-batch checkpoint" (Damian et al., 2021), which is the 100% training accuracy point reached by running full-batch GD on the training loss function. The result is plotted as Figure 8 in Subsection B.1. One can observe that USAM still gets stuck at the "full-batch checkpoint", while SAM keeps increasing its test accuracy via drifting along the minima manifold.

## 4 Case Study: Learning Threshold Neurons for Sparse Coding Problem

To incorporate our two findings into a single example, we consider training one-hidden-layer ReLU networks for the sparse coding problem, a setup considered in (Ahn et al., 2023a) to study the role of $\eta$ in GD. Without going into details, the crux of their experiment is to understand how GD with large $\eta$ finds desired structures of the network – in this specific case, the desired structure is the negative bias in ReLU unit (also widely known as "thresholding unit/neuron"). In this section, we evaluate SAM and USAM under the same setup, illustrating the importance of normalization.

**Main observation of Ahn et al. (2023a).** Given this background, the main observation of Ahn et al. (2023a) is that i) when training the ReLU network with GD, different learning rates induce very different trajectories; and ii) the desired structure, namely a ***negative bias in ReLU, only arises with large "unstable" learning rates*** for which GD exhibits unstable behaviors. We reproduce their results in Figure 6, plotting the test accuracy on the left and the bias of ReLU unit on the right. As they claimed, GD with larger $\eta$ learns more negative bias, which leads to better test accuracy.

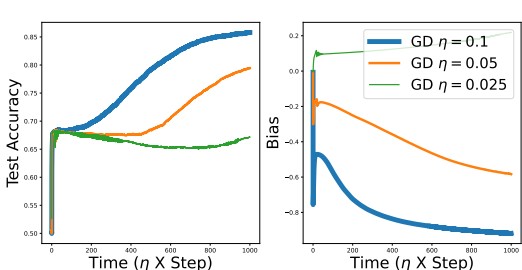

Figure 6: Behavior of GD for sparse coding problem.

Their inspiring observation is however a bit discouraging for practitioners. According to their theoretical results, such learning rates have to be quite large – large to the point where GD shows very unstable behavior (à la Edge-of-Stability (Cohen et al., 2021)). In practice, without knowledge of the problem, this requires a careful hyper-parameter search to figure out the correct learning rate. More importantly, such large and unstable learning rates may cause GD to diverge or lead to worse performance. More discussions can be found in the recent paper by Kaur et al. (2022).

In contrast, as we will justify shortly, ***SAM does not suffer from such a "dilemma of tuning"*** – matching with our results in Theorem 4. Moreover, ***the removal of normalization no longer attains such a property***, as we demonstrated in Theorem 5. In particular, for USAM, one also needs to carefully tune $\eta$ and $\rho$ for better performance – as we inspired in Theorem 5 and Theorem 6, small $(\eta, \rho)$ makes the iterates get stuck early; on the other hand, as we presented in Section 2, an aggressive choice causes USAM to diverge. The following experiments illustrate these claims in more detail.

In Figure 7, we plot the performance of SAM, USAM, and GD with different $\eta$'s (while fixing $\rho$) – gray lines for GD, solid lines for SAM, and dashed lines for USAM. From the plot, USAM behaves more similarly to GD than SAM: the bias does not decrease sufficiently when the learning rate is not large enough, which consequently to leads poor test accuracy. On the other hand, no matter what

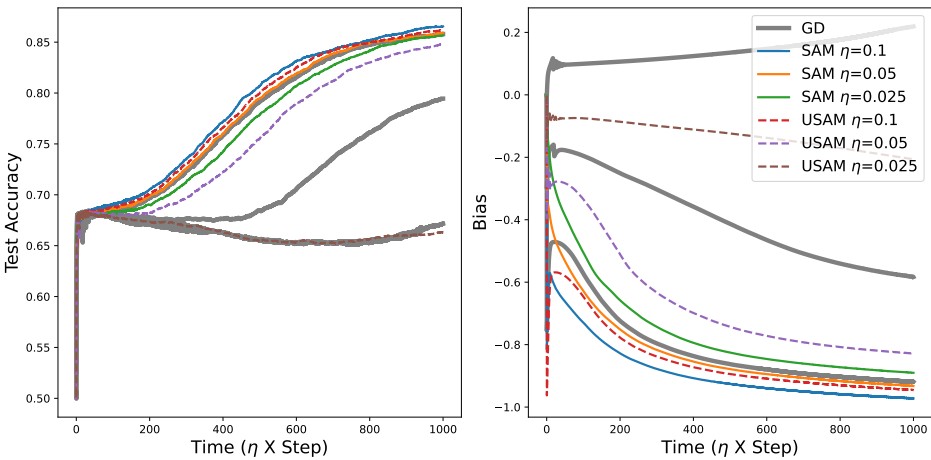

Figure 7: Behaviors of different algorithms for sparse coding problem ($\rho = 0.01$). The gray curves (corresponding to GD with different learning rates) are taken from Figure 6 with the same set of $\eta$'s.

$\eta$ is chosen for SAM, bias is negative enough and ensures better generalization. Hence, Figure 7 illustrates that compared to SAM, USAM is less robust to the tuning of $\eta$.

In Figure 9 (deferred to Subsection B.2), we also compare these three algorithms when varying $\rho$ and fixing $\eta$. In addition to what we observe in Figure 7, we show that normalization also helps stability – USAM quickly diverges as we increase $\rho$, while SAM remains robust to the choice of $\rho$. Thus, USAM is also less robust to the tuning of $\rho$. In other words, our observation in Figure 7 extends to $\rho$.

Hence, putting Figure 7 and Figure 9 together, we conclude that ***SAM is robust to different configurations of*** $(\eta, \rho)$ ***while USAM is robust to neither of them***. Hence, the normalization of SAM eases hyper-parameter tuning, which is typically a tough problem for GD and many other algorithms – normalization boosts the success of SAM in practice.

## 5 Conclusion

In this paper, we investigate the role played by normalization in SAM. By theoretically characterizing the behavior of SAM and USAM on both convex and non-convex losses and empirically verifying our conclusions via real-world neural networks, we found that normalization i) helps stabilize the algorithm iterates, and ii) enables the algorithm to keep moving along the manifold of minimizers, leading to better performance in many cases. Moreover, as we demonstrate via various experiments, these two properties make SAM require less hyper-parameter tuning, supporting its practicality.

In this work, we follow a recent research paradigm of "physics-style" approaches to understanding deep neural networks based on simplified models (c.f. (Zhang et al., 2022; Garg et al., 2022; von Oswald et al., 2023; Abernethy et al., 2023; Allen-Zhu and Li, 2023; Liu et al., 2023; Li et al., 2023; Ahn et al., 2023a,b,c)). We found such physics-style approaches quite helpful, especially for complex modern neural networks. We hope that our work builds stepping stones for future works on understanding working mechanisms of modern deep neural networks.

## Acknowledgments

Kwangjun Ahn was supported by the ONR grant (N00014-20-1-2394) and MIT-IBM Watson as well as a Vannevar Bush fellowship from Office of the Secretary of Defense. Suvrit Sra acknowledges support from an NSF CAREER grant (1846088), and NSF CCF-2112665 (TILOS AI Research Institute). Kwangjun Ahn also acknowledges support from the Kwanjeong Educational Foundation. We thank Kaiyue Wen, Zhiyuan Li, and Hadi Daneshmand for their insightful discussions.

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

# Appendix

# A    Setup of the Motivating Experiment

In the motivating experiments (Figure 1 and Figure 2), we follow the over-parameterized matrix sensing setup as Li et al. (2018) and Blanc et al. (2020). Specifically, we do the following:

1. Generate the true matrix by sampling each entry of $U^\star \in \mathbb{R}^{d \times r}$ independently from a standard Gaussian distribution and let $X^\star = U^\star (U^\star)^\top$.
2. Normalize each column of $U^\star$ to unit norm so that the spectral norm of $U^\star$ is close to one.
3. For every sensing matrix $A_i$ ($i = 1, 2, \ldots, m$), sample the entries of $A_i$ independently from a standard Gaussian distribution. Then observe $b_i = \langle A_i, X^\star \rangle$.

In particular, for the experiments, we chose $r = 5$, $d = 100$, and $m = 5dr$.

# B    Additional Experimental Results

## B.1    Running SAM and USAM from Other Initializations

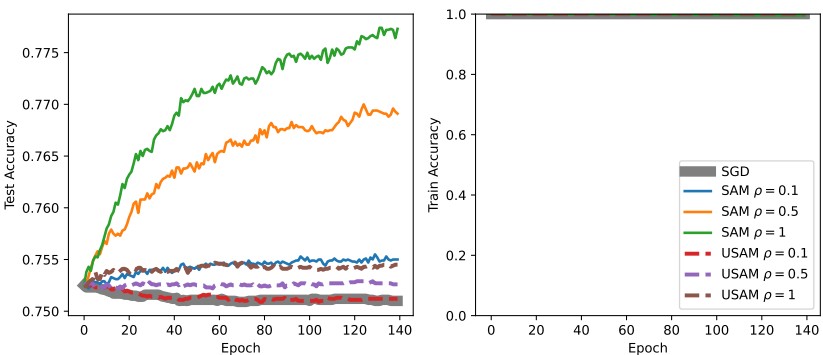

Figure 8: Training ResNet18 on CIFAR-10 from a "full-batch checkpoint" of Damian et al. (2021) ($\eta = 0.001$, batch size $= 128$).

## B.2    Varying $\rho$ While Fixing $\eta$ in Sparse Coding Example

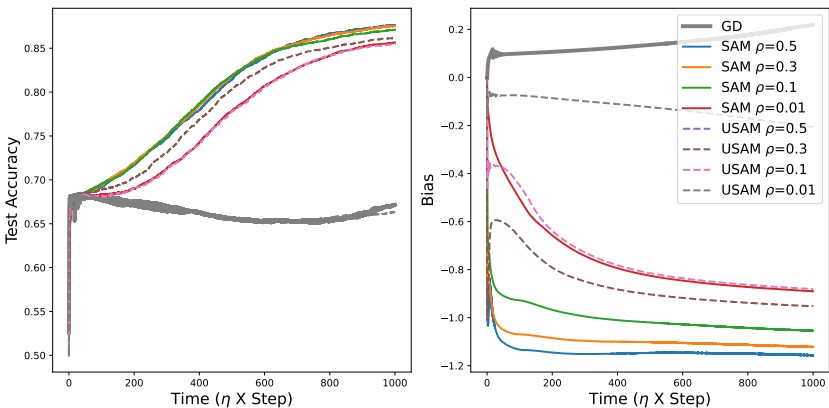

Figure 9: Behaviors of different algorithms for sparse coding problem ($\eta = 0.025$). Note that USAM with $\rho = 0.5$ (the dashed violet curve) diverges and becomes invisible except for the very beginning.

Aside from Figure 7 which varies $\eta$ while fixing $\rho$, we perform the same experiment when varying $\rho$ and fixing $\eta$. The main observation for SAM is similar to that of Figure 7: different hyper-parameters all keep decreasing the bias and give better test accuracy – even with the tiniest choice $\rho = 0.01$.

However, for USAM, there are three different types of $\rho$'s as shown in Figure 9:

1. For tiny $\rho = 0.01$, the bias doesn't decrease much. Consequently, the performance of USAM nearly degenerates to that of GD – while SAM with $\rho = 0.01$ still gives outstanding performance.
2. For moderate $\rho = 0.1$ and $\rho = 0.3$, USAM manages to decrease the bias and improve its accuracy, though with a slower speed than SAM with the same choices of $\rho$.
3. For large $\rho = 0.5$ (where SAM still works well; see the solid curve in blue), USAM diverges.

Thus, the dilemma described in Section 4 indeed applies to not only $\eta$ but also $\rho$ – matching our main conclusion that normalization helps make hyper-parameter tuning much more manageable.

## C  Omitted Proof for Smooth and Strongly Convex Losses

We shall first restate Theorem 1 here for the ease of presentation.

**Theorem 7** (Restatement of Theorem 1). *For any $\alpha$-strongly-convex and $\beta$-smooth loss function $\mathcal{L}$, for any learning rate $\eta \leq {}^2/_\beta$ and perturbation radius $\rho \geq 0$, the following holds:*

1. **SAM.** *The iterate $w_t$ converges to a local neighborhood around the minimizer $w^\star$. Formally,*

$$\mathcal{L}(w_t) - \mathcal{L}(w^\star) \leq \left(1 - \alpha\eta(2 - \eta\beta)\right)^t (\mathcal{L}(w_0) - \mathcal{L}(w^\star)) + \frac{\eta\beta^3\rho^2}{2\alpha(2 - \eta\beta)}, \quad \forall t.$$

2. **USAM.** *In contrast, there exists some $\alpha$-strongly-convex and $\beta$-smooth loss $\mathcal{L}$ such that the USAM with $\eta \in \left({}^2/_{(\beta+\rho\beta^2)}, {}^2/_\beta\right]$ diverges for all except measure zero initialization $w_0$.*

We will show these two conclusions separately. The proof directly follows from Theorem 8 and Theorem 10.

### C.1  SAM Allows a Descent Lemma Like GD

**Theorem 8** (SAM over Strongly Convex and Smooth Losses). *For any $\alpha$-strongly-convex and $\beta$-smooth loss function $\mathcal{L}$, for any learning rate $\eta \leq {}^2/_\beta$ and perturbation radius $\rho \geq 0$, the following holds for SAM:*

$$\mathcal{L}(w_t) - \mathcal{L}(w^\star) \leq (1 - \alpha\eta(2 - \eta\beta))^t (\mathcal{L}(w_0) - \mathcal{L}(w^\star)) + \frac{\eta\beta^3\rho^2}{2\alpha(2 - \eta\beta)}.$$

*Proof.* We first claim the following analog of descent lemma, which we state as Lemma 9:

$$\mathcal{L}(w_{t+1}) \leq \mathcal{L}(w_t) - \frac{1}{2}\eta(2 - \eta\beta)\|\nabla\mathcal{L}(w_t)\|^2 + \frac{\eta^2\beta^3\rho^2}{2}.$$

By definition of strong convexity, we have

$$\mathcal{L}(w_t) - \mathcal{L}(w^\star) \leq \langle \nabla\mathcal{L}(w_t), w_t - w^\star \rangle - \frac{\alpha}{2}\|w_t - w^\star\|^2 \leq \frac{1}{2\alpha}\|\nabla f(w_t)\|^2,$$

where the last inequality uses the fact that $\langle a, b \rangle - \frac{1}{2}\|b\|^2 \leq \frac{1}{2}\|a\|^2$. Thus, combining the two inequalities above, we obtain

$$\mathcal{L}(w_{t+1}) \leq \mathcal{L}(w_t) - \alpha\eta(2 - \eta\beta)(\mathcal{L}(w_t) - \mathcal{L}(w^\star)) + \frac{\eta^2\beta^3\rho^2}{2},$$

which after rearrangement becomes

$$\mathcal{L}(w_{t+1}) - \mathcal{L}(w^\star) \leq (1 - \alpha\eta(2 - \eta\beta))(\mathcal{L}(w_t) - \mathcal{L}(w^\star)) + \frac{\eta^2\beta^3\rho^2}{2}.$$

Unrolling this recursion, we obtain

$$\mathcal{L}(w_t) - \mathcal{L}(w^\star) \leq (1 - \alpha\eta(2 - \eta\beta))^t (\mathcal{L}(w_0) - \mathcal{L}(w^\star)) + \frac{\eta^2\beta^3\rho^2}{2}\sum_{k=0}^{t-1}(1 - \alpha\eta(2 - \eta\beta))^k$$

$$\leq (1 - \alpha\eta(2 - \eta\beta))^t (\mathcal{L}(w_0) - \mathcal{L}(w^\star)) + \frac{\eta^2\beta^3\rho^2}{2} \sum_{k=0}^{\infty} (1 - \alpha\eta(2 - \eta\beta))^k$$

$$= (1 - \alpha\eta(2 - \eta\beta))^t (\mathcal{L}(w_0) - \mathcal{L}(w^\star)) + \frac{\eta^2\beta^3\rho^2}{2\alpha\eta(2 - \eta\beta)} \,.$$

This completes the proof. □

**Lemma 9** (SAM Descent lemma). *For a convex loss $\mathcal{L}$ that is $\beta$-smooth, SAM iterates $w_t$ satisfy the following when the learning rate $\eta < \frac{2}{\beta}$ and $\rho \geq 0$:*

$$\mathcal{L}(w_{t+1}) \leq \mathcal{L}(w_t) - \frac{1}{2}\eta(2 - \eta\beta)\|\nabla\mathcal{L}(w_t)\|^2 + \frac{\eta^2\beta^3\rho^2}{2}, \quad \forall t.$$

*Proof.* Let $v_t \triangleq \nabla\mathcal{L}(w_t)/\|\nabla\mathcal{L}(w_t)\|$ and $w_{t+1/2} \triangleq w_t + \rho v_t$ so we have $w_{t+1} = w_t - \eta\nabla\mathcal{L}(w_{t+1/2})$. Then the $\beta$-smoothness of $\mathcal{L}$ yields

$$\mathcal{L}(w_{t+1}) \leq \mathcal{L}(w_t) - \eta\langle\nabla\mathcal{L}(w_t), \nabla\mathcal{L}(w_{t+1/2})\rangle + \frac{\eta^2\beta}{2}\|\nabla\mathcal{L}(w_{t+1/2})\|^2.$$

We start with upper bounding the norm of $\nabla\mathcal{L}(w_{t+1/2})$:

$$\|\nabla\mathcal{L}(w_{t+1/2})\|^2 = \|\nabla\mathcal{L}(w_{t+1/2}) - \nabla\mathcal{L}(w_t)\|^2 - \|\nabla\mathcal{L}(w_t)\|^2 + 2\langle\nabla\mathcal{L}(w_{t+1/2}), \nabla\mathcal{L}(w_t)\rangle$$
$$\leq \beta^2\rho^2 - \|\nabla\mathcal{L}(w_t)\|^2 + 2\langle\nabla\mathcal{L}(w_{t+1/2}), \nabla\mathcal{L}(w_t)\rangle,$$

Hence, as long as $\eta < \frac{2}{\beta}$, we have the following upper bound on $\mathcal{L}(w_{t+1})$:

$$\mathcal{L}(w_{t+1}) \leq \mathcal{L}(w_t) - \eta\langle\nabla\mathcal{L}(w_{t+1/2}), \nabla\mathcal{L}(w_t)\rangle + \frac{\eta^2\beta}{2}\|\nabla\mathcal{L}(w_{t+1/2})\|^2$$

$$\leq \mathcal{L}(w_t) - \eta\langle\nabla\mathcal{L}(w_{t+1/2}), \nabla\mathcal{L}(w_t)\rangle + \frac{\eta^2\beta}{2}\left(\beta\rho^2 - \|\nabla\mathcal{L}(w_t)\|^2 + 2\langle\nabla\mathcal{L}(w_{t+1/2}), \nabla\mathcal{L}(w_t)\rangle\right)$$

$$= \mathcal{L}(w_t) - \frac{\eta^2\beta}{2}\|\nabla\mathcal{L}(w_t)\|^2 - (\eta - \eta^2\beta)\langle\nabla\mathcal{L}(w_{t+1/2}), \nabla\mathcal{L}(w_t)\rangle + \frac{\eta^2\beta^3\rho^2}{2}.$$

Now we lower bound $\langle\nabla\mathcal{L}(w_{t+1/2}), \nabla\mathcal{L}(w_t)\rangle$. Note that

$$\langle\nabla\mathcal{L}(w_t + \rho v_t), \nabla\mathcal{L}(w_t)\rangle = \langle\nabla\mathcal{L}(w_t + \rho v_t) - \nabla\mathcal{L}(w_t), \nabla\mathcal{L}(w_t)\rangle + \|\nabla\mathcal{L}(w_t)\|^2$$
$$= \frac{\|\nabla\mathcal{L}(w_t)\|}{\rho}\langle\nabla\mathcal{L}(w_t + \rho v_t) - \nabla\mathcal{L}(w_t), \rho v_t\rangle + \|\nabla\mathcal{L}(w_t)\|^2 \geq \|\nabla\mathcal{L}(w_t)\|^2,$$

where the last inequality uses the following standard fact about convex functions: for any $w_1, w_2$, $\langle\nabla\mathcal{L}(w_1) - \nabla\mathcal{L}(w_2), w_1 - w_2\rangle \geq 0$. Hence, we arrive at

$$\mathcal{L}(w_{t+1}) \leq \mathcal{L}(w_t) - \frac{\eta^2\beta}{2}\|\nabla\mathcal{L}(w_t)\|^2 - (\eta - \eta^2\beta)\langle\nabla\mathcal{L}(w_{t+1/2}), \nabla\mathcal{L}(w_t)\rangle + \frac{\eta^2\beta^3\rho^2}{2}$$

$$\leq \mathcal{L}(w_t) - \frac{\eta^2\beta}{2}\|\nabla\mathcal{L}(w_t)\|^2 - (\eta - \eta^2\beta)\|\nabla\mathcal{L}(w_t)\|^2 + \frac{\eta^2\beta^3\rho^2}{2}$$

$$\leq \mathcal{L}(w_t) - \frac{1}{2}\eta(2 - \eta\beta)\|\nabla\mathcal{L}(w_t)\|^2 + \frac{\eta^2\beta^3\rho^2}{2},$$

and thus finishing the proof. □

### C.2 USAM Diverges on Quadratic Losses

**Theorem 10** (USAM over Quadratic Losses). *Following (Bartlett et al., 2022), consider the quadratic loss $\mathcal{L}$ induced by a PSD matrix $\Lambda$. Without loss of generality, let $\mathcal{L}$ minimize at the origin. Formally,*

$$\mathcal{L}(w) = \frac{1}{2}w^\top\Lambda w, \quad \text{where } \Lambda = \text{diag}(\lambda_1, \ldots, \lambda_d), \tag{8}$$

*where $\lambda_{\max} = \lambda_1 > \cdots \geq \lambda_d > 0$ and $v_{\max} = e_1, e_2, \ldots, e_d$ are the eigenvectors corresponding to $\lambda_1, \lambda_2, \ldots, \lambda_d$, respectively. Then the iterates of USAM Equation 2 applied to Equation 8 satisfy:*

1. *If $\eta(\lambda_{\max} + \rho\lambda_{\max}^2) < 2$, then the iterates converges to the global minima exponentially fast.*
2. *If $\eta(\lambda_{\max} + \rho\lambda_{\max}^2) > 2$ and if $\langle w_0, v_{\max} \rangle \neq 0$, then the iterates diverge.*

*Moreover, such a loss function $\mathcal{L}$ is $\lambda_1$-smooth and $\lambda_d$-strongly-convex.*

*Proof.* As $\nabla\ell(w) = \Lambda w$, the USAM update Equation 2 reads

$$w_{t+1} = w_t - \eta\nabla\mathcal{L}(w_t + \rho\nabla\mathcal{L}(w_t)) = (I - \eta\Lambda - \eta\rho\Lambda^2)w_t.$$

Hence, if $\eta(\lambda_{\max} + \rho\lambda_{\max}^2) < 2$, then since $\|I - \eta\Lambda - \eta\rho\Lambda^2\| < 1$, we have that $\|w_t\| \to 0$ exponentially fast. On the other hand, if $\eta(\lambda_{\max} + \rho\lambda_{\max}^2) > 2$, then $|1 - \eta\lambda_1 - \eta\lambda_1^2| > 1$. Since $\|w_t\| \geq |1 - \eta\lambda_1 - \eta\lambda_1^2|^t|\langle w_0, e_1 \rangle|$, it follows that the iterates diverge as long as $\langle w_0, v_{\max} \rangle \neq 0$. Finally, as $\nabla^2\mathcal{L}(w) = \Lambda$, we directly know that $\mathcal{L}$ is $\lambda_1$-smooth and $\lambda_d$-strongly-convex. $\square$

## D    Omitted Proof for Scalar Factorization Problems

**Theorem 11** (Formal Version of Theorem 2). *Consider the scalar factorization loss $\mathcal{L}(x, y) = \frac{1}{2}(xy)^2$. If $(x_0, y_0)$ satisfies $2x_0 > y_0 > x_0 \gg 0$ and $\eta = (x_0^2 + y_0^2)^{-1}$, then i) SAM finds a neighborhood of the origin with radius $\mathcal{O}(\rho)$ for all $\rho$, and ii) USAM diverges as long as $\rho \geq 15\eta$.*

This theorem is a combination of Theorem 12 and Theorem 15 that we will show shortly.

### D.1    SAM Always Converges on Scalar Factorization Problems

Let $w_t = \begin{bmatrix} x_t & y_t \end{bmatrix}^\top$ be the $t$-th iterate. Denote $\nabla_t = \nabla\mathcal{L}(w_t) = \begin{bmatrix} x_t y_t^2 & x_t^2 y_t \end{bmatrix}^\top$. For each step $t$, the actual update of SAM is therefore the gradient taken at

$$\tilde{w}_t = w_t + \rho\frac{\nabla_t}{\|\nabla_t\|} = \begin{bmatrix} x_t + \rho y_t z_t^{-1} \\ y_t + \rho x_t z_t^{-1} \end{bmatrix}, \quad \text{where } z_t^{-1} = \frac{\text{sign}(x_t y_t)}{\sqrt{x_t^2 + y_t^2}}.$$

By denoting $\tilde{\nabla}_t = \nabla\mathcal{L}(\tilde{w}_t)$, the update rule of SAM is

$$\begin{bmatrix} x_{t+1} \\ y_{t+1} \end{bmatrix} = w_{t+1} = w_t - \eta\tilde{\nabla}_t = \begin{bmatrix} x_t - \eta(x_t + \rho y_t z_t^{-1})(y_t + \rho x_t z_t^{-1})^2 \\ y_t - \eta(x_t + \rho y_t z_t^{-1})^2(y_t + \rho x_t z_t^{-1}) \end{bmatrix}.$$

We make the following claim:

**Theorem 12** (SAM over Scalar Factorization Problems). *Under the setting of Theorem 11, there exists some threshold $T$ for SAM such that $|x_t|, |y_t| \leq 5\rho$ for all $t \geq T$.*

*Proof.* We first observe that SAM over $\mathcal{L}(x, y)$ always pushes the iterate towards the minima (which are all the points on the $x$- and $y$- axes). Formally:

- If $x_t \geq 0$, then $x_{t+1} \leq x_t$. If $x_t \leq 0$, then $x_{t+1} \geq x_t$.
- If $y_t \geq 0$, then $y_{t+1} \leq y_t$. If $y_t \leq 0$, then $y_{t+1} \geq y_t$.

This observation can be verified by noticing $\text{sign}(\rho y_t z_t^{-1}) = \text{sign}(y_t)\text{sign}(x_t y_t) = \text{sign}(x_t)$ and similarly $\text{sign}(\rho x_t z_t^{-1}) = \text{sign}(y_t)$. In other words, $\text{sign}(x_t + \rho y_t z_t^{-1}) = \text{sign}(x_t)$ and thus always pushes $x_t$ towards the $y$-axis. The same also holds for $y_t$ by symmetry.

In analog to the descent lemma for GD, we can show the following lemma:

**Lemma 13.** *When picking $\eta = (x_0^2 + y_0^2)^{-1}$, we have $\mathcal{L}(w_{t+1}) - \mathcal{L}(w_t) < 0$ as long as $x_t, y_t \geq 5\rho$.*

*Proof.* As $\nabla^2\mathcal{L}(x, y) = \begin{bmatrix} y^2 & 2xy \\ 2xy & x^2 \end{bmatrix}$, we know $\mathcal{L}$ is $\beta \triangleq (x_0^2 + y_0^2)$-smooth inside the region $\{(x, y) : x^2 + y^2 \leq \beta\}$. Then we have (recall that $\eta = (x_0^2 + y_0^2)^{-1} = 1/\beta$)

$$\mathcal{L}(w_{t+1}) - \mathcal{L}(w_t) \leq \langle \nabla\mathcal{L}(w_t), w_{t+1} - w_t \rangle + \frac{\beta}{2}\|w_{t+1} - w_t\|^2$$

$$= -\eta\langle\nabla_t, \tilde{\nabla}_t\rangle + \frac{\beta}{2}\eta^2\|\tilde{\nabla}_t\|^2 = -\eta\Big(\langle\nabla_t - \tfrac{1}{2}\tilde{\nabla}_t, \tilde{\nabla}_t\rangle\Big).$$

To make sure that it's negative, we simply want

$$0 \le \langle\nabla_t - \tfrac{1}{2}\tilde{\nabla}_t, \tilde{\nabla}_t\rangle = \left(x_t y_t^2 - \frac{1}{2}\big(x_t + \rho y_t z_t^{-1}\big)\big(y_t + \rho x_t z_t^{-1}\big)^2\right)\big(x_t + \rho y_t z_t^{-1}\big)\big(y_t + \rho x_t z_t^{-1}\big)^2 +$$

$$\left(x_t^2 y_t - \frac{1}{2}\big(x_t + \rho y_t z_t^{-1}\big)^2\big(y_t + \rho x_t z_t^{-1}\big)\right)\big(x_t + \rho y_t z_t^{-1}\big)^2\big(y_t + \rho x_t z_t^{-1}\big),$$

which can be ensured once

$$\big(x_t + \rho y_t z_t^{-1}\big)\big(y_t + \rho x_t z_t^{-1}\big)^2 \le 2x_t y_t^2, \quad \big(x_t + \rho y_t z_t^{-1}\big)^2\big(y_t + \rho x_t z_t^{-1}\big) \le 2x_t^2 y_t.$$

If we have $x_t, y_t \ge 5\rho$, then as $z_t^{-1} \le \min\{x_t^{-1}, y_t^{-1}\}$, we have

$$\big(x_t + \rho y_t z_t^{-1}\big)\big(y_t + \rho x_t z_t^{-1}\big)^2 \le (x_t + \rho)(y_t + \rho)^2 \le 1.2^3 x_t y_t^2 < 2x_t y_t^2,$$

which shows the first inequality. The second one follows from symmetry. $\qquad\square$

Therefore, SAM will progress until $x_t \le 5\rho$ or $y_t \le 5\rho$. Without loss of generality, assume that $x_t \le 5\rho$; we then claim that $y_t$ will soon decrease to $\mathcal{O}(\rho)$.

**Lemma 14.** *Suppose that $|x_t| \le 5\rho$ but $y_t \ge 5\rho$. Then $|x_{t+1}| \le 5\rho$ but $y_{t+1} \le (1 - \frac{1}{2}\eta\rho^2)y_t$.*

*Proof.* First, show that $|x_t|$ remains bounded by $5\rho$. Assume $x_t \ge 0$ without loss of generality:

$$\begin{aligned}
x_{t+1} &= x_t - \eta\big(x_t + \rho y_t z_t^{-1}\big)\big(y_t + \rho x_t z_t^{-1}\big)^2 \\
&\ge x_t - \eta(5\rho + \rho)(y_t + \rho)^2 \\
&\ge -\eta 6\rho\left(\frac{4}{5\eta} + \rho^2 + 2\rho\sqrt{\frac{4}{5\eta}}\right) \ge -5\rho,
\end{aligned} \tag{9}$$

where the second last line uses $y_t^2 \le y_0^2 = \frac{y_0^2}{x_0^2 + y_0^2}\eta^{-1} \le \frac{4}{5}\eta^{-1}$ and the last one uses $\eta = (x_0^2 + y_0^2)^{-1} \ll 1$. Meanwhile, we see that $y_t$ decreases exponentially fast by observing the following:

$$\begin{aligned}
y_{t+1} &= y_t - \eta(y_t + \rho x_t z_t^{-1})(x_t + \rho y_t z_t^{-1})^2 \\
&\le y_t - \eta y_t(\rho y_t z_t^{-1})^2 \\
&\le \left(1 - \frac{1}{2}\eta\rho^2\right)y_t,
\end{aligned}$$

where the last line uses $z_t^{-1} = (x_t^2 + y_t^2)^{-1/2} \ge (2y_t^2)^{-1/2} = 2^{-1/2}y_t^{-1}$ as $y_t \ge 5\rho \ge |x_t|$. $\qquad\square$

So eventually we have $|x_t|, |y_t| \le 5\rho$. Recall that Equation 9 infers $|x_{t+1}| \le 5\rho$ from $|x_t| \le 5\rho$. Hence, by symmetry, we conclude that $|x_{t+1}|, |y_{t+1}| \le 5\rho$ hold as well. Therefore, SAM always finds an $\mathcal{O}(\rho)$-neighborhood of the origin, i.e., it is guaranteed to converge regardless of $\rho$. $\qquad\square$

### D.2 USAM Diverges with Small $\rho$

For USAM, the dynamics can be written as

$$\begin{bmatrix} x_{t+1} \\ y_{t+1} \end{bmatrix} = \begin{bmatrix} x_t \\ y_t \end{bmatrix} - \eta\nabla\mathcal{L}\begin{pmatrix} x_t + \rho x_t y_t^2 \\ y_t + \rho x_t^2 y_t \end{pmatrix} = \begin{bmatrix} x_t - \eta(x_t + \rho x_t y_t^2)(y_t + \rho x_t^2 y_t)^2 \\ y_t - \eta(x_t + \rho x_t y_t^2)^2(y_t + \rho x_t^2 y_t) \end{bmatrix}. \tag{10}$$

We make the following claim which is similar to Theorem 10:

**Theorem 15** (USAM over Scalar Factorization Problems)**.** *Under the setup of Theorem 11, for any $\rho \ge 15\eta$, $|x_{t+1}| \ge 2|x_t|$ and $|y_{t+1}| \ge 2|y_t|$ for all $t \ge 1$; in other words, USAM diverges exponentially fast.*

*Proof.* Prove by induction. From Equation 10, our conclusion follows once

$$\eta\big|(x_t + \rho x_t y_t^2)(y_t + \rho x_t^2 y_t)^2\big| \geq 3|x_t|, \quad \eta\big|(x_t + \rho x_t y_t^2)^2(y_t + \rho x_t^2 y_t)\big| \geq 3|y_t|.$$

According to our setup that $\eta = (x_0^2 + y_0^2)^{-1}$, $y_0 \leq 2x_0$, and the induction statement that, we have $\eta \geq (5x_t^2)^{-1}$. The second inequality then holds as long as

$$\big|(\rho x_t y_t^2)^2(\rho x_t^2 y_t)\big| \geq 15x_t^2|y_t|, \quad \text{i.e., } \rho^3 x_t^2 y_t^4 \geq 15,$$

which is true as $x_t^2 \geq x_0^2$, $y_t^4 \geq y_0^4$, $y_0 \geq x_0$, and $\rho \geq 15\eta \geq 3x_0^{-2}$. Note that the bounds on $\rho$ are very loose, and we made no effort to optimize it; instead, we only aimed to show that USAM starts to diverge from a $\rho = \Theta(\eta) \ll 1$. $\qquad \square$

# E   Assumptions in the Single-Neuron Linear Network Model

**Assumptions on $\ell$.** Following Ahn et al. (2023a), we make the following assumptions about $\ell$:

(A1)  $\ell$ is a convex, even, 1-Lipschitz function that is minimized at 0.
(A2)  $\ell$ is twice continuously differentiable near the origin with $\ell''(0) = 1$, which infers the existence of a constant $c > 0$ such that $|\ell'(s)| \leq |s|$ for all $|s| \leq c$.
(A3)  We further assume a "linear tail" away from the minima, i.e., $|\ell'(s)| \geq c/2$ for all $|s| \geq c$ and $|\ell'(s)| \geq |s|/2$ for $|s| \leq c$.

Some concrete example of loss functions satisfying the above assumption include a symmetrized logistic loss $\frac{1}{2}\log(1 + \exp(-2s)) + \frac{1}{2}\log(1 + \exp(2s))$ and a square root loss $\sqrt{1 + s^2}$ One may refer to their paper for more details.

# F   Omitted Proof of SAM Over Single-Neuron Linear Networks

**Theorem 16** (Formal Version of Theorem 4)**.** *For a loss $\mathcal{L}$ over $(x, y) \in \mathbb{R}^2$ defined as $\mathcal{L}(x, y) = \ell(xy)$ where $\ell$ satisfies Assumptions (A1), (A2), and (A3), if the initialization $(x_0, y_0)$ satisfies:*

$$y_0 \geq x_0 \gg 0, \quad y_0^2 - x_0^2 = \frac{\gamma}{\eta}, \quad y_0^2 = C\frac{\gamma}{\eta},$$

*where $\gamma \in [\frac{1}{2}, 2]$ and $C \geq 1$ is constant, then for all hyper-parameter configurations $(\eta, \rho)$ such that[4]*

$$\eta\rho + \sqrt{C\gamma\eta} \leq \min\left\{\frac{1}{2}, C\right\}\sqrt{\frac{\gamma}{\eta}}, \quad \frac{4\sqrt{C}}{c}\eta^{-1} = \mathcal{O}(\min\{\eta^{-1.5}\rho^{-1}\gamma^{1/2}, \eta^{-2}\}),$$

*we can decompose the trajectory of SAM (defined in Equation 1) into three phases, whose main conclusions are stated separately in three theorems and are informally summarized here:*

1. *(Theorem 20) Until $x_t = \mathcal{O}(\sqrt{\gamma\eta})$, we must have $y_t = \Omega(\sqrt{\gamma/\eta})$, and $x_{t+1} \leq x_t - \Omega(\sqrt{\gamma\eta})$.*
2. *(Theorem 26) After Initial Phase and until $y_t \leq |x_t|$, $|x_t| = \mathcal{O}(\eta\rho + \sqrt{\eta})$ still holds. Meanwhile, $y_{t+1} \leq y_t - \min\{\Omega(\eta\rho^2)y_t, \Omega(\eta\rho)\}$ (i.e., $y_t$ either drops by $\Omega(\eta\rho)$ or decays by $\Omega(\eta\rho^2)$).*
3. *(Theorem 32) After Middle Phase, we always have $|x_t|, |y_t| = \mathcal{O}(\eta\rho + \sqrt{\eta})$.*

## F.1   Basic Properties and Notations

Recall the update rule of SAM in Equation 1: $w_{t+1} \leftarrow w_t - \eta\nabla\mathcal{L}(w_t + \rho\frac{\nabla\mathcal{L}(w_t)}{\|\nabla\mathcal{L}(w_t)\|})$. By writing $w_t$ as $\begin{bmatrix} x_t \\ y_t \end{bmatrix}$, substituting $\mathcal{L}(x, y) = \ell(xy)$, and utilizing the expressions of $\nabla\mathcal{L}(x, y)$ in Equation 6, we have:

$$w_t + \rho\frac{\nabla\mathcal{L}(w_t)}{\|\nabla\mathcal{L}(w_t)\|} = \begin{bmatrix} x_t \\ y_t \end{bmatrix} + \rho\frac{\ell'(x_t y_t)}{|\ell'(x_t y_t)|\sqrt{x_t^2 + y_t^2}}\begin{bmatrix} y_t \\ x_t \end{bmatrix} = \begin{bmatrix} x_t \\ y_t \end{bmatrix} + \rho\frac{\text{sign}(x_t y_t)}{\sqrt{x_t^2 + y_t^2}}\begin{bmatrix} y_t \\ x_t \end{bmatrix}, \quad (11)$$

---

[4]As $y_0 \gg 0$, we must have $\eta \ll 1$; thus, these conditions automatically hold when $\eta = o(1)$ and $\rho = \mathcal{O}(1)$.

where the second step uses the fact that $\ell$ is a even function so $\ell'(t)$ has the same sign with $t$. Define $z_t = \frac{\sqrt{x_t^2 + y_t^2}}{\text{sign}(x_t y_t)}$. Then $w_t + \rho \frac{\nabla \mathcal{L}(w_t)}{\|\nabla \mathcal{L}(w_t)\|} = \begin{bmatrix} x_t + \rho y_t z_t^{-1} \\ y_t + \rho x_t z_t^{-1} \end{bmatrix}$. Further denoting $\ell'\big((x_t + \rho y_t z_t^{-1})(y_t + \rho x_t z_t^{-1})\big)$ by $\ell'_t$, one can simplify the update rule Equation 1 as follows:

$$
\begin{aligned}
w_{t+1} &= \begin{bmatrix} x_t \\ y_t \end{bmatrix} - \eta \ell'\big((x_t + \rho y_t z_t^{-1})(y_t + \rho x_t z_t^{-1})\big) \begin{bmatrix} y_t + \rho x_t z_t^{-1} \\ x_t + \rho y_t z_t^{-1} \end{bmatrix} \\
&= \begin{bmatrix} x_t \\ y_t \end{bmatrix} - \eta \ell'_t \begin{bmatrix} y_t + \rho x_t z_t^{-1} \\ x_t + \rho y_t z_t^{-1} \end{bmatrix}.
\end{aligned}
\tag{12}
$$

**Assumption.** Following footnote 1, we assume that $x_t, y_t \neq 0$ for all $t$.

We are ready to give some basic properties of Equation 12. First, we claim that the sign of $\ell'_t$ is the same as the product $x_t y_t$. Formally, we have the following lemma:

**Lemma 17** (Sign of Gradient in SAM). *If $x_t \neq 0, y_t \neq 0$, then $\text{sign}(x_t) = \text{sign}(x_t + \rho y_t z_t^{-1})$ and $\text{sign}(y_t) = \text{sign}(y_t + \rho x_t z_t^{-1})$. In particular, if $x_t \neq 0, y_t \neq 0$, we must have $\text{sign}(\ell'_t) = \text{sign}(x_t y_t)$.*

*Proof.* Note that $\text{sign}(z_t) = \text{sign}(x_t y_t)$, so $\text{sign}(y_t z_t^{-1}) = \text{sign}(x_t)$. Similarly, $\text{sign}(x_t z_t^{-1}) = \text{sign}(y_t)$. Thus, we have $\text{sign}(x_t) = \text{sign}(x_t + \rho y_t z_t^{-1})$ and $\text{sign}(y_t) = \text{sign}(y_t + \rho x_t z_t^{-1})$, which in turn shows that $\text{sign}(\ell'_t) = \text{sign}\big(\text{sign}(x_t + \rho y_t z_t^{-1}) \, \text{sign}(y_t + \rho x_t z_t^{-1})\big) = \text{sign}(x_t y_t)$. $\qquad \square$

Using Lemma 17, the SAM update can be equivalently written as

$$
\begin{bmatrix} x_{t+1} \\ y_{t+1} \end{bmatrix} = \begin{bmatrix} x_t \\ y_t \end{bmatrix} - \eta \ell'_t \begin{bmatrix} y_t \\ x_t \end{bmatrix} - \eta |\ell'_t| \begin{bmatrix} \rho x_t (x_t^2 + y_t^2)^{-1/2} \\ \rho y_t (x_t^2 + y_t^2)^{-1/2} \end{bmatrix}.
\tag{13}
$$

We then claim the following property, which can be intuitively interpolated as SAM always goes towards the minimizes (i.e., both axes) for each step, although sometimes it may overshoot.

**Lemma 18.** *Suppose that $x_t, y_t \neq 0$. Then, the following basic properties hold:*

- *If $x_t > 0$, then $x_{t+1} < x_t$. If $x_t < 0$, then $x_{t+1} > x_t$.*
- *If $y_t > 0$, then $y_{t+1} < y_t$. If $y_t < 0$, then $y_{t+1} > y_t$.*

*Proof.* We only show the case that $x_t > 0$ as the other cases are similar. Recall the dynamics of $x_t$:

$$
x_{t+1} = x_t - \eta \ell'_t \cdot \rho z_t^{-1} x_t - \eta \ell'_t \cdot y_t .
$$

Since $x_t \neq 0$, Lemma 17 implies that $\text{sign}(\ell'_t) = \text{sign}(x_t y_t)$, and so

$$
\begin{aligned}
x_{t+1} &= x_t - \eta \ell'_t \cdot \rho z_t^{-1} x_t - \eta \ell'_t \cdot y_t \\
&= x_t - \eta \, \text{sign}(x_t y_t) |\ell'_t| \cdot \rho \, \text{sign}(x_t y_t)(x_t^2 + y_t^2)^{-1/2} x_t - \eta \, \text{sign}(x_t y_t) |\ell'_t| \cdot \text{sign}(y_t) |y_t| \\
&= x_t - \eta |\ell'_t| \cdot \rho (x_t^2 + y_t^2)^{-1/2} x_t - \eta |\ell'_t| \cdot |y_t| \leq x_t,
\end{aligned}
$$

where the last line uses the assumption that $x_t > 0$. $\qquad \square$

## F.2 Initial Phase: $x_t$ Decreases Fast while $y_t$ Remains Large

As advertised in Theorem 16, we group all rounds until $x_t \leq \frac{c}{4}\sqrt{\gamma\eta}$ as Initial Phase. Formally:

**Definition 19** (Initial Phase). *Let $t_1$ be the largest time such that $x_t > \frac{c}{4}\sqrt{\gamma\eta}$ for all $t \leq t_1$. We call the iterations $[0, t_1]$ **Initial Phase**.*

**Theorem 20** (Main Conclusion of Initial Phase; SAM Case). *For $t_0 = \Theta(\min\{\eta^{-1.5}\rho^{-1}\gamma^{1/2}, \eta^{-2}\})$, we have $y_t \geq \frac{1}{2}\sqrt{\gamma/\eta}$ for all $t \leq \min\{t_0, t_1\}$ under the conditions of Theorem 16. Moreover, we have $x_{t+1} \leq x_t - \frac{c}{4}\sqrt{\gamma\eta}$ for all $t \leq \min\{t_0, t_1\}$, which consequently infers $t_1 \leq t_0$ under the conditions of Theorem 16, i.e., $\min\{t_0, t_1\}$ is just $t_1$. This shows the first claim of Theorem 16.*

**Remark.** This theorem can be intuitively explained as follows: At initialization, $x_0, y_0$ are both $\Theta(\sqrt{1/\eta})$. However, after $t_1$ iterations, we have $|x_{t_1+1}| = \mathcal{O}(\sqrt{\eta})$; meanwhile, $y_t$ is still of order $\Omega(\sqrt{1/\eta})$ (much larger than $\mathcal{O}(\sqrt{\eta})$). Hence, $|x_t|$ and $|y_t|$ get widely separated in Initial Phase.

*Proof.* This theorem is a direct consequence of Lemma 21 and Lemma 24. □

**Lemma 21.** *There exists $t_0 = \Theta(\min\{\eta^{-1.5}\rho^{-1}\gamma^{1/2}, \eta^{-2}\})$ such that $|y_t| \geq \frac{1}{2}\sqrt{\gamma/\eta}$ for $t \leq t_0$. If $\eta$ is sufficiently small s.t. $\eta\rho + \sqrt{C\gamma\eta} \leq \frac{1}{2}\sqrt{\gamma/\eta}$, then $y_t \geq \frac{1}{2}\sqrt{\gamma/\eta}$ for $t \leq \min\{t_1, t_0\}$.*

*Proof.* Let $t_0$ be defined as follows:

$$t_0 \triangleq \max\left\{ t \ : \ \left(1 - \frac{8}{\sqrt{C}}\eta^{1.5}\gamma^{-1/2}\rho - \eta^2\right)^t \geq \frac{1}{4}\right\}. \tag{14}$$

Then, it follows that $t_0 = \Theta(\min\{\eta^{-1.5}\rho^{-1}\gamma^{1/2}, \eta^{-2}\})$. We first prove the following claim.

**Claim 22.** *For all $t \leq t_0$, $y_t^2 \geq \frac{1}{4}\gamma/\eta$.*

*Proof.* We prove this claim by induction. We trivially have $y_0^2 = C\gamma/\eta \geq \frac{1}{4}C\gamma/\eta$. Now suppose that the conclusion holds up to some $t < t_0$, i.e., $y_{t'}^2 \geq \frac{1}{4}\gamma/\eta$ for all $t' \leq t$. We consider $y_{t+1}^2 - x_{t+1}^2$:

$$
\begin{aligned}
y_{t+1}^2 - x_{t+1}^2 &= (1 - \eta\ell_t' \cdot \rho z_t^{-1})^2(y_t^2 - x_t^2) - (\eta\ell_t')^2(y_t^2 - x_t^2) \\
&= \left(1 - 2\eta\ell_t' \cdot \rho z_t^{-1} + \eta^2(\ell_t' \cdot \rho z_t^{-1})^2 - \eta^2(\ell_t')^2\right) \cdot (y_t^2 - x_t^2) \\
&\overset{(a)}{\geq} (1 - 2\eta\rho z_t^{-1} - \eta^2) \cdot (y_t^2 - x_t^2) \\
&\geq (1 - 2\eta\rho y_t^{-1} - \eta^2) \cdot (y_t^2 - x_t^2) \\
&\overset{(b)}{\geq} \left(1 - \frac{8}{\sqrt{C}}\eta^{1.5}\gamma^{-1/2}\rho - \eta^2\right) \cdot (y_t^2 - x_t^2),
\end{aligned}
$$

where (a) used $\ell_t' \leq 1$ (thanks to Assumption (A1)) and (b) used the induction hypothesis $y_t^2 \geq \frac{1}{4}C\gamma/\eta$. This further implies that

$$y_{t+1}^2 - x_{t+1}^2 \geq \left(1 - \frac{8}{\sqrt{C}}\eta^{1.5}\gamma^{-1/2}\rho - \eta^2\right)^{t+1}(y_0^2 - x_0^2) = \left(1 - \frac{8}{\sqrt{C}}\eta^{1.5}\gamma^{-1/2}\rho - \eta^2\right)^{t+1}\frac{\gamma}{\eta}.$$

Thus, by the definition of $t_0$, we must have $y_{t+1}^2 \geq y_{t+1}^2 - x_{t+1}^2 \geq \frac{1}{4}\gamma/\eta$, which proves the claim. □

Next, we prove the second conclusion.

**Claim 23.** *If $\eta\rho + \sqrt{C\gamma\eta} \leq \frac{1}{2}\sqrt{\gamma/\eta}$, then $y_t \geq \frac{1}{2}\sqrt{\gamma/\eta}$ for $t \leq \min\{t_0, t_1\}$.*

*Proof.* Still show by induction. Let $y_t \geq \frac{1}{2}\sqrt{\gamma/\eta} > 0$ for some $t < \min\{t_0, t_1\}$. Consider $y_{t+1}$.
By Definition 19, $x_t$ is positive for all $t \leq t_1$. Thus, using Lemma 18, we have $x_t \leq x_0 \leq y_0 = \sqrt{C\gamma/\eta}$. Since $x_t, y_t > 0$, we have $\text{sign}(\ell_t') = \text{sign}(x_t y_t) = \text{sign}(x_t) > 0$ and hence (13) gives

$$
\begin{aligned}
y_{t+1} &= y_t - \eta|\ell_t'|\rho(x_t^2 + y_t^2)^{-1/2}|y_t| - \eta|\ell_t'||x_t| \\
&\geq y_t - \eta\rho - \eta|x_t|,
\end{aligned}
$$

where we used the facts that $|\ell_t'| \leq 1$ and $(x_t^2 + y_t^2)^{-1/2} \leq \min\{|x_t|^{-1}, |y_t|^{-1}\}$. Hence, we get

$$y_{t+1} \geq y_t - \eta\rho - \sqrt{C\gamma\eta} \geq 0$$

since $\eta\rho + \sqrt{C\gamma\eta} \leq \frac{1}{2}\sqrt{\gamma/\eta} \leq y_t$. By Claim 22, $y_{t+1} \geq 0$ implies $y_{t+1} \geq \frac{1}{2}\sqrt{\gamma/\eta}$ as well. □

Combining Claim 22 and Claim 23 finishes the proof of Lemma 21. □

**Lemma 24.** *For any $t \leq \min\{t_0, t_1\}$, we have*

$$x_{t+1} \leq x_t - \frac{c}{4}\sqrt{\gamma\eta}.$$

*In particular, if $\eta$ is sufficiently small s.t. $\frac{4\sqrt{C}}{c}\eta^{-1} \leq t_0$, then we must have $t_1 \leq \frac{4}{c}\eta^{-1} - 1 < t_0$.*

*Proof.* Since $x_t > 0$ for all $t \leq t_1$, we have $\text{sign}(\ell'_t) = \text{sign}(x_t y_t) = \text{sign}(y_t)$ from Lemma 17 and Lemma 21, so the SAM update (13) becomes

$$x_{t+1} = x_t - \eta |\ell'_t| \cdot \rho(x_t^2 + y_t^2)^{-1/2} x_t - \eta |\ell'_t| \cdot |y_t|.$$

Since $x_t > 2c\sqrt{\frac{\eta}{C\gamma}}$, we have

$$(x_t + \rho y_t z_t^{-1})(y_t + \rho x_t z_t^{-1}) > x_t y_t \geq 2c\sqrt{\frac{\eta}{\gamma}} \cdot \frac{1}{2}\sqrt{\frac{\gamma}{\eta}} \geq c,$$

which implies that $\ell'_t \geq c/2$ from Assumption (A3). Together with Lemma 21, we have

$$x_{t+1} = x_t - \eta |\ell'_t| \cdot \rho(x_t^2 + y_t^2)^{-1/2} z_t^{-1} x_t - \eta |\ell'_t| \cdot |y_t|$$
$$\leq x_t - \eta |\ell'_t| \cdot |y_t| \leq x_t - \eta \frac{c}{2} \frac{1}{2}\sqrt{\frac{\gamma}{\eta}} = x_t - \frac{c}{4}\sqrt{\gamma\eta}.$$

Let $t'_1 \triangleq \frac{\sqrt{C\gamma/\eta}}{\frac{c}{4}\sqrt{\gamma\eta}} = \frac{4\sqrt{C}}{c}\eta^{-1}$. Since $x_0 \leq \sqrt{C\gamma/\eta}$, we have $x_{t'_1} < \frac{c}{4}\sqrt{\gamma\eta}$ as long as $t_1 \leq t_0$. Thus, it follows that $t_1 \leq t'_1 - 1 < t_0$, as desired. $\qquad\square$

### F.3 Middle Phase: $y$ Keeps Decreasing Until Smaller Than $|x_t|$

Then we move on to the second claim of Theorem 16. We define all rounds until $y_t < |x_t|$ that are after Initial Phase as Middle Phase. Formally, we have the following definition.

**Definition 25** (Middle Phase). *Let $t_2$ be the first time that $y_t < |x_t|$. We call $(t_1, t_2)$ **Middle Phase**.*

Before presenting the main conclusion of Middle Phase, we first define a threshold $B$ that we measure whether a variable is "small enough", which we will use throughout this section. Formally,

$$B \triangleq \max\left\{\frac{2}{c}\sqrt{\frac{\eta}{C\gamma}}, \ \eta\rho + \sqrt{C\eta\gamma}\right\}.$$

**Theorem 26** (Main Conclusion of Middle Phase; SAM Case). *Under the conditions of Theorem 16, $|y_t| \leq \sqrt{C\gamma/\eta}$ and $|x_t| \leq B$ throughout Middle Phase. Under the same conditions, we further have $y_{t+1} \leq y_t - \min\{\frac{1}{2}\eta\rho^2 y_t, \frac{c}{2\sqrt{2}}\eta\rho\}$ for all $t_1 \leq t \leq t_2$ – showing the second claim of Theorem 16.*

**Remark.** This theorem can be understood as follows. Upon entering Middle Phase, $|x_t|$ is bounded by $\mathcal{O}(\eta\rho + \sqrt{\eta})$. This then gets preserved throughout Middle Phase. Meanwhile, in $t_2$ iterations, $y_t$ drops rapidly such that $y_{t_2+1} \leq |x_{t_2+1}| = \mathcal{O}(\eta\rho + \sqrt{\eta})$. In other words, $x_t$ and $y_t$ are both "close enough" to 0 after Middle Phase; thus, SAM finds the flattest minimum (which is the origin).

*Proof.* The claims are shown in Lemma 27, Lemma 28 and Lemma 30, respectively. $\qquad\square$

**Lemma 27.** *If $\eta\rho + \sqrt{C\gamma\eta} < \sqrt{C\gamma/\eta}$, then during the Middle Phase, $|y_t| \leq y_0 \leq \sqrt{C\gamma/\eta}$.*

*Proof.* When $\text{sign}(y_{t+1}) = \text{sign}(y_t)$, then Lemma 18 implies that $|y_{t+1}| \leq |y_t|$. Now consider the case $\text{sign}(y_{t+1}) = -\text{sign}(y_t)$. For simplicity, say $y_t > 0$ and $y_{t+1} < 0$. Since $y_t > 0$, we have $\text{sign}(\ell'_t) = \text{sign}(x_t y_t) = \text{sign}(x_t)$ and hence Equation 13 gives

$$y_{t+1} = y_t - \eta|\ell'_t|\rho(x_t^2 + y_t^2)^{-1/2}|y_t| - \eta|\ell'_t||x_t|$$
$$\geq -\eta\rho - \eta|x_t| \geq -\eta\rho - \eta y_t$$
$$\geq -\eta\rho - \eta\sqrt{\frac{C\gamma}{\eta}} = -\eta\rho - \sqrt{C\gamma\eta},$$

which shows that $|y_{t+1}| \leq \eta\rho + \sqrt{C\gamma\eta}$. This proves the statement. $\qquad\square$

**Lemma 28.** *Suppose that $|x_t| \leq B = \max\left\{\frac{2}{c}\sqrt{\frac{\eta}{C\gamma}}, \ \eta\rho + \sqrt{C\eta\gamma}\right\}$. Then we have $|x_{t+1}| \leq B$.*

*Proof.* By symmetry, we only consider the case where $x_t > 0$. If $x_{t+1} \geq 0$, then we have $0 \leq x_{t+1} \leq x_t$ due to Lemma 24. If $x_{t+1} < 0$, then since $x_t > 0$, it follows that

$$x_{t+1} = \text{sign}(x_t)x_{t+1} = |x_t| - \eta|\ell_t'|\rho(x_t^2 + y_t^2)^{-1/2}|x_t| - \eta|\ell_t'||y_t|$$
$$\geq |x_t| - \eta\rho - \eta\sqrt{C\gamma/\eta} \geq -\eta\rho - \sqrt{C\eta\gamma}, \tag{15}$$

where the last inequality is because $|y_t| \leq y_0 \leq \sqrt{C\gamma/\eta}$. This concludes the proof. $\qquad\square$

**Corollary 29.** *Note that, by definition of Initial Phase in Definition 19, we already have $|x_{t_1+1}| \leq B$. Hence, this lemma essentially says $|x_t| \leq B = \max\left\{\frac{2}{c}\sqrt{\frac{\eta}{C\gamma}}, \eta\rho + \sqrt{C\eta\gamma}\right\}$ for all $t_1 < t \leq t_2$.*

**Lemma 30.** *If $y_t \geq |x_t|$, then we must have*

$$y_{t+1} \leq y_t - \min\left\{\frac{1}{2}\eta\rho^2 y_t, \ \frac{c}{2\sqrt{2}}\eta\rho\right\}.$$

*Proof.* Since $y_t \geq |x_t|$, we have $(x_t^2 + y_t^2)^{-1/2} \geq (2y_t^2)^{-1/2} = (\sqrt{2}y_t)^{-1}$. Consider two cases:

i) if $|(x_t + \rho y_t z_t^{-1})(y_t + \rho x_t z_t^{-1})| \geq c$, then $|\ell_t'| \in [\frac{c}{2}, 1]$ according to Assumption (A3). Moreover, since $\text{sign}(\ell_t') = \text{sign}(x_t y_t) = \text{sign}(x_t)$, it follows that

$$y_{t+1} = y_t - \eta|\ell_t'| \cdot \rho(x_t^2 + y_t^2)^{-1/2}y_t - \eta\ell_t' \cdot x_t$$
$$= \left(1 - \eta|\ell_t'|\rho(x_t^2 + y_t^2)^{-1/2}\right)y_t - \eta|\ell_t'||x_t|$$
$$\leq \left(1 - \eta c_t\rho(\sqrt{2}y_t)^{-1}\right)y_t$$
$$\leq y_t - \eta\rho\frac{c}{2}\frac{1}{\sqrt{2}} = y_t - \Omega(\eta\rho). \tag{16}$$

ii) otherwise, it follows from (A3) that

$$|\ell_t'| \geq \frac{1}{2}|(x_t + \rho y_t z_t^{-1})(y_t + \rho x_t z_t^{-1})|.$$

As $\text{sign}(x_t) = \text{sign}(\rho y_t z_t^{-1})$ and $\text{sign}(y_t) = \text{sign}(\rho x_t z_t^{-1})$, it follows that

$$|\ell_t'| \cdot \rho z_t^{-1}y_t \geq \frac{1}{2}|(x_t + \rho y_t z_t^{-1})(y_t + \rho x_t z_t^{-1})| \cdot \rho z_t^{-1}y_t$$
$$\geq \frac{1}{2}(\rho y_t z_t^{-1})^2 y_t$$

we must have

$$y_{t+1} \leq y_t - \frac{1}{2}\eta(\rho y_t z_t^{-1})^2(y_t) \leq y_t - \frac{1}{2}\eta\rho^2 y_t = (1 - \Omega(\eta\rho^2))y_t, \tag{17}$$

where we used $z_t^{-1} \geq (\sqrt{2}y_t)^{-1}$ and thus $y_t z_t^{-1} \geq 1/\sqrt{2}$.

Combining item 16 and item 17, we obtain the desired conclusion. $\qquad\square$

## F.4  Final Phase: Both $x_t$ and $y_t$ Oscillates Around the Origin

It only remains to show that the iteration never escapes the origin.

**Definition 31** (Final Phase). *We denote by "**Final Phase**" all iterations after $t_2$.*

**Theorem 32** (Main Conclusion of Final Phase; SAM Case). *For all $t > t_2$, we have $|x_t|, |y_t| \leq B$ where $B = \max\left\{\frac{2}{c}\sqrt{\frac{\eta}{C\gamma}}, \eta\rho + \sqrt{C\eta\gamma}\right\}$.*

**Remark.** As we will see shortly, when entering Final Phase, both $|x_t|$ and $|y_t|$ are bounded by $\mathcal{O}(\eta\rho + \sqrt{\eta})$. This theorem essentially says that they can never exceed this bound in Final Phase. In other words, in Final Phase, both $x_t$ and $y_t$ are oscillating around 0.

*Proof.* We first show the following lemma, ruling out the possibility of $|y_t| > B$ after Middle Phase:

**Lemma 33.** *If $\eta\rho + \eta B \leq B$,[5] then $|y_{t_2+1}| \leq B = \max\left\{\frac{2}{c}\sqrt{\frac{\eta}{C\gamma}},\ \eta\rho + \sqrt{C\eta\gamma}\right\}$.*

*Proof.* The proof will be very similar to Lemma 28. Recall that by Definition 25, we must have $y_{t_2} > 0$. If $y_{t_2+1} > 0$ as well, then $|y_{t_2+1}| = y_{t_2+1} \leq |x_{t_2+1}| \leq B$. Otherwise,

$$y_{t_2+1} = \text{sign}(y_{t_2})y_{t_2+1} = |y_{t_2}| - \eta|\ell'_{t_2}|\rho(x_{t_2}^2 + y_{t_2}^2)^{-1/2}|y_{t_2}| - \eta|\ell'_{t_2}||x_{t_2}| \geq -\eta\rho - \eta B,$$

where we used $|x_{t_2}| \leq B$. We proved our claim as $\eta\rho + \eta B \leq B$. $\qquad\square$

According to Lemma 28 Lemma 33, we have

$$|x_{t_2+1}|, |y_{t_2+1}| \leq B = \max\left\{\frac{2}{c}\sqrt{\frac{\eta}{C\gamma}},\ \eta\rho + \sqrt{C\eta\gamma}\right\}. \tag{18}$$

Hence, it only remains to do a reduction, stated as follows.

**Lemma 34.** *If $\eta\rho + \eta B \leq B$, then $|x_{t+1}|, |y_{t+1}| \leq B$ as long as $|x_t|, |y_t| \leq B$.*

*Proof.* Without loss of generality, let $x_t > 0$ for some $t > t_2$ and show that $|x_{t+1}| \leq B$ – which is identical to the proof of Lemma 27. By symmetry, the same also holds for $|x_t|$. $\qquad\square$

Our conclusion thus follows from an induction based on Equation 18 and Lemma 34. $\qquad\square$

# G   Omitted Proof of USAM Over Single-Neuron Linear Networks

**Theorem 35** (Formal Version of Theorem 5). *For a loss $\mathcal{L}$ over $(x,y) \in \mathbb{R}^2$ defined as $\mathcal{L}(x,y) = \ell(xy)$ where $\ell$ satisfies Assumptions (A1), (A2), and (A3), if the initialization $(x_0, y_0)$ satisfies:*

$$y_0 \geq x_0 \gg 0, \quad y_0^2 - x_0^2 = \frac{\gamma}{\eta}, \quad y_0^2 = C\frac{\gamma}{\eta},$$

*where $\gamma \in [\frac{1}{2}, 2]$ and $C \geq 1$ is constant, then for all hyper-parameter configurations $(\eta, \rho)$ such that[6]*

$$\eta \leq \frac{1}{2}, \quad \eta\rho \leq \min\left\{\frac{1}{2}, C^{-1}\gamma^{-1}\right\}, \quad 2\sqrt{C}\eta^{-1} = \mathcal{O}(\min\{\eta^{-1}\rho^{-1}, \eta^{-2}\}), \quad C\gamma\left(1 + \rho C\frac{\gamma}{\eta}\right) \geq 16,$$

*we can characterize the initial and final phases of the trajectory of USAM (defined in Equation 2) by the following two theorems, whose main conclusions are informally summarized below:*

1. *(**Theorem 39**) Until $x_t = \mathcal{O}(\sqrt{\gamma\eta})$, we must have $y_t = \Omega(\sqrt{\gamma/\eta})$, and $x_{t+1} \leq x_t - \Omega(\sqrt{\gamma\eta})$.*
2. *(**Theorem 44**) Once $(1 + \rho y_t^2)y_t^2 \lesssim 2/\eta$, $|x_t|$ decays exponentially and thus USAM gets stuck.*

Different from Theorem 16, there is no characterization of Middle Phase here, which means the iterates can also stop above the threshold $\tilde{y}^2_{\text{USAM}}$ defined in Equation 7 (the technical reason is sketched in subsubsection 3.1.3, i.e., SAM gradients non non-negligible when $y_t$ is large, while USAM gradients vanish once $|x_t|$ is small). However, we remark that the main takeaway of Theorem 35 is to contrast SAM (which always attains $y_\infty^2 \approx 0$) with USAM (which must stop once $(1 + \rho y_t^2)y_t^2 \lesssim 2/\eta$).

## G.1   Basic Properties and Notations

Recall the update rule of USAM in Equation 2: $w_{t+1} \leftarrow w_t - \eta\nabla\mathcal{L}(w_t + \rho\nabla\mathcal{L}(w_t))$. Still writing $w_t$ as $[x_t \quad y_t]^\top$, we have

$$w_{t+1} = \begin{bmatrix} x_t \\ y_t \end{bmatrix} - \eta\nabla\mathcal{L}\left(\begin{bmatrix} x_t \\ y_t \end{bmatrix} + \rho\ell'(x_t y_t)\begin{bmatrix} y_t \\ x_t \end{bmatrix}\right)$$

---

[5]Recall that $B = \max\{\frac{2}{c}\sqrt{\frac{\eta}{C\gamma}}, \eta\rho + \sqrt{C\eta\gamma}\}$, we only need to ensure that $\eta B \leq \sqrt{C\eta\gamma}$, which can be done by $\eta \leq \min\{cC\gamma, \frac{1}{4}C\gamma\rho^{-2}, \frac{1}{2}\}$. As the RHS is of order $\Omega(1)$, $\eta = \mathcal{O}(1)$ again suffices.

[6]Similar to Theorem 16, these conditions are satisfied once $\eta = o(1)$ and $\rho = \mathcal{O}(1)$.

$$= \begin{bmatrix} x_t \\ y_t \end{bmatrix} - \eta \ell' \big( (x_t + \rho\ell'(x_t y_t)y_t)(y_t + \rho\ell'(x_t y_t)x_t) \big) \begin{bmatrix} y_t + \rho\ell'(x_t y_t)x_t \\ x_t + \rho\ell'(x_t y_t)y_t \end{bmatrix}. \tag{19}$$

Due to the removal of normalization, $\ell'$ are taken twice at different points in Equation 19. For simplicity, we denote $\tilde{\ell}'_t = \ell'(x_t y_t)$ and $\ell'_t = \ell'((x_t + \rho\tilde{\ell}'_t y_t)(y_t + \rho\tilde{\ell}'_t x_t))$. The update rule can be rewritten as:

$$x_{t+1} = x_t - \eta\ell'_t \cdot (y_t + \rho\tilde{\ell}'_t x_t), \quad y_{t+1} = y_t - \eta\ell'_t \cdot (x_t + \rho\tilde{\ell}'_t y_t). \tag{20}$$

Similar to the SAM case, we shall approximate $\tilde{\ell}'_t$ and $\ell'_t$ according to the magnitude of $x_t y_t$ and $(x_t + \rho\tilde{\ell}'_t y_t)(y_t + \rho\tilde{\ell}'_t x_t)$. We first have the following lemma similar to Lemma 17:

**Lemma 36** (Sign of Gradient in USAM). *If $x_t \neq 0, y_t \neq 0$, then $\mathrm{sign}(x_t) = \mathrm{sign}(x_t + \rho\tilde{\ell}'_t y_t)$ and $\mathrm{sign}(y_t) = \mathrm{sign}(y_t + \rho\tilde{\ell}'_t x_t)$. In particular, if $x_t \neq 0, y_t \neq 0$, then $\mathrm{sign}(\ell'_t) = \mathrm{sign}(\tilde{\ell}'_t) = \mathrm{sign}(x_t y_t)$.*

*Proof.* First of all, $\mathrm{sign}(\tilde{\ell}'_t) = \mathrm{sign}(x_t y_t)$ according to Assumption (A1). Therefore,

$$\mathrm{sign}(x_t + \rho\tilde{\ell}'_t y_t) = \mathrm{sign}(x_t + \rho\,\mathrm{sign}(x_t y_t)y_t) = \mathrm{sign}(x_t),$$
$$\mathrm{sign}(y_t + \rho\tilde{\ell}'_t x_t) = \mathrm{sign}(y_t + \rho\,\mathrm{sign}(x_t y_t)x_t) = \mathrm{sign}(y_t),$$

giving our first two claims. The last conclusion follows by definition. □

Moreover, we also have the following lemma analog to Lemma 18:

**Lemma 37.** *Suppose that $x_t, y_t \geq 0$. Then, the following basic properties hold:*

- *If $x_t > 0$, then $x_{t+1} < x_t$. If $x_t < 0$, then $x_{t+1} > x_t$.*
- *If $y_t > 0$, then $y_{t+1} < y_t$. If $y_t < 0$, then $y_{t+1} > y_t$.*

*Proof.* We only prove the statement for $x_t > 0$ as the proof is similar for other cases. Recall the dynamics of $x_t$:

$$x_{t+1} = x_t - \eta\ell'_t \cdot (y_t + \rho\tilde{\ell}'_t x_t).$$

Since $x_t \neq 0$, Lemma 36 implies that $\mathrm{sign}(\ell'_t) = \mathrm{sign}(\tilde{\ell}'_t) = \mathrm{sign}(x_t y_t)$, and so

$$\begin{aligned}
x_{t+1} &= x_t - \eta\ell'_t \cdot (y_t + \rho\tilde{\ell}'_t x_t) \\
&= x_t - \eta\,\mathrm{sign}(x_t y_t)|\ell'_t| \cdot (y_t + \rho\,\mathrm{sign}(x_t y_t)|\tilde{\ell}'_t|x_t) \\
&= x_t - \eta|\ell'_t| \cdot (|y_t| + \rho|\tilde{\ell}'_t|x_t) \leq x_t,
\end{aligned}$$

where the last line uses the assumption that $x_t > 0$. □

### G.2 Initial Phase: $x_t$ Decreases Fast while $y_t$ Remains Large

The definition of Initial Phase is very similar to the one of SAM – and the conclusion is also analogue, although the proofs are slightly different because of the different update rule.

**Definition 38** (Initial Phase). *Let $t_1$ be the largest time such that $x_t > \frac{1}{2}\sqrt{\gamma\eta}$ for all $t \leq t_1$. We denote by "**Initial Phase**" the iterations $[0, t_1]$.*

**Theorem 39** (Main Conclusion of Initial Phase; USAM Case). *For $t_0 = \Theta(\min\{\eta^{-1}\rho^{-1}, \eta^{-2}\})$, we have $y_t \geq \frac{1}{2}\sqrt{\gamma/\eta}$ for all $t \leq \min\{t_0, t_1\}$ under the conditions of Theorem 35. Moreover, we have $x_{t+1} \leq x_t - \frac{1}{2}\sqrt{C\gamma\eta}$ for all $t \leq \min\{t_0, t_1\}$, which consequently infers $t_1 \leq t_0$ under the conditions of Theorem 35, i.e., $\min\{t_0, t_1\}$ is just $t_1$. This shows the first claim of Theorem 35.*

*Proof.* This theorem is a combination of Lemma 40 and Lemma 43. □

Similar to Lemma 21, $y_t$ in USAM also cannot be too small in the first few iterations:

**Lemma 40.** *There exists $t_0 = \Theta(\min\{\eta^{-1}\rho^{-1}, \eta^{-2}\})$ such that for $t \leq t_0$, we have $|y_t| \geq \frac{1}{2}\sqrt{\gamma/\eta}$. Assuming $\eta \leq \frac{1}{2}$, then $y_t \geq \frac{1}{2}\sqrt{\gamma/\eta}$ for $t \leq \min\{t_1, t_0\}$.*

*Proof.* The proof idea follows from Lemma 21. Let $t_0$ be defined as follows:

$$t_0 \triangleq \max\left\{ t \ : \ \left(1 - \eta^2 - 2\eta\rho\right)^t \geq \frac{1}{4} \right\}. \tag{21}$$

Then, it follows that $t_0 = \Theta(\min\{\eta^{-1}\rho^{-1}, \eta^{-2}\})$. We first show that $y_t^2$ cannot be too small until $t_0$:

**Claim 41.** For all $t \leq t_0$, $y_t^2 \geq \frac{1}{4}\gamma/\eta$.

*Proof.* Prove by induction. Initially, $y_0^2 = C\gamma/\eta \geq \frac{1}{4}\gamma/\eta$. Then consider some $t < t_0$ such that $y_{t'}^2 \geq \frac{1}{4}\gamma/\eta$ for all $t' \leq t$. By Equation 20, we have

$$
\begin{aligned}
y_{t+1}^2 - x_{t+1}^2 &= \left(y_t - \eta\ell_t' \cdot (x_t + \rho\tilde{\ell}_t' y_t)\right)^2 - \left(x_t - \eta\ell_t' \cdot (y_t + \rho\tilde{\ell}_t' x_t)\right)^2 \\
&= (y_t^2 - x_t^2) + (\eta\ell_t')^2\left((x_t + \rho\tilde{\ell}_t' y_t)^2 - (y_t + \rho\tilde{\ell}_t' x_t)^2\right) + \\
&\quad\ 2x_t \cdot \eta\ell_t' \cdot (y_t + \rho\tilde{\ell}_t' x_t) - 2y_t \cdot \eta\ell_t' \cdot (x_t + \rho\tilde{\ell}_t' y_t) \\
&= (y_t^2 - x_t^2) + (\eta\ell_t')^2(x_t^2 - y_t^2) + (\eta\ell_t')^2(\rho\tilde{\ell}_t')^2(y_t^2 - x_t^2) + 2\eta\ell_t' \cdot \rho\tilde{\ell}_t'(x_t^2 - y_t^2) \\
&= \left(1 - (\eta\ell_t')^2 + (\eta\ell_t')^2(\rho\tilde{\ell}_t')^2 - 2\eta\ell_t' \cdot \rho\tilde{\ell}_t'\right)(y_t^2 - x_t^2).
\end{aligned}
$$

Using Assumption (A1), we know that $|\ell_t'| \leq 1$ and $|\tilde{\ell}_t'| \leq 1$, giving

$$y_{t+1}^2 - x_{t+1}^2 \geq \left(1 - \eta^2 - 2\eta\rho\right)(y_t^2 - x_t^2) \geq \cdots \geq \left(1 - \eta^2 - 2\eta\rho\right)^{t+1}(y_0^2 - x_0^2).$$

By definition of $t_0$ and condition that $t < t_0$, we have $y_{t+1}^2 \geq y_{t+1}^2 - x_{t+1}^2 \geq \frac{1}{4}\gamma/\eta$. $\qquad\square$

**Claim 42.** Suppose that $\sqrt{C\gamma\eta} \leq \frac{1}{2}\sqrt{\gamma/\eta}$, i.e., $\eta \leq \frac{1}{2}\sqrt{C}$. Then $y_t \geq \frac{1}{2}\sqrt{\gamma/\eta}$ for all $t \leq t_0$.

*Proof.* We still consider the maximum single-step difference in $y_t$. Suppose that for some $t' < \min\{t_0, t_1\}$, $y_{t'} \geq \frac{1}{2}\sqrt{\gamma/\eta}$ for all $t' \leq t$. According to Equation 20 and Assumption (A1),

$$y_{t+1} = y_t - \eta\ell_t' \cdot (x_t + \rho\tilde{\ell}_t' y_t) \geq y_t - \eta(x_t + \rho y_t) \geq (1 - \eta\rho)y_t - \eta\sqrt{C\gamma/\eta},$$

where the last inequality used Lemma 37 to conclude that $x_t \leq x_0 \leq y_0 = \sqrt{C\gamma/\eta}$. Hence, as the first term is non-negative and $\sqrt{C\gamma\eta} \leq \frac{1}{2}\sqrt{\gamma/\eta}$, we must have $y_{t+1} \geq 0$, which implies $y_{t+1} \geq \frac{1}{2}\sqrt{\gamma/\eta}$ according to Claim 41. $\qquad\square$

Putting these two claims together gives our conclusion. Note that the condition $\eta \leq \frac{1}{2}\sqrt{C}$ in Claim 42 can be inferred from the assumptions $\eta \leq \frac{1}{2}$ and $C \geq 1$ made in Theorem 35. $\qquad\square$

After showing that $y_t$ never becomes too small, we are ready to show that $x_t$ decreases fast enough.

**Lemma 43.** *For $t \leq \min\{t_0, t_1\}$, we have*

$$x_{t+1} \leq x_t - \frac{1}{2}\sqrt{\gamma\eta}.$$

*In particular, if $\eta$ is sufficiently small s.t. $2\sqrt{C}\eta^{-1} \leq t_0$, then we must have $t_1 \leq 2\sqrt{C}\eta^{-1} - 1 < t_0$.*

*Proof.* Since $x_t > 0$ for all $t \leq t_1$, we have $\text{sign}(\ell_t') = \text{sign}(x_t y_t) = \text{sign}(y_t)$ from Lemma 36 Lemma 40, so the USAM update (20) simplifies as follows according to Assumption (A1) and Lemma 40

$$x_{t+1} = x_t - \eta|\ell_t'| \cdot (|y_t| + \rho|\ell_t'||x_t|) \leq x_t - \eta|y_t| \leq x_t - \frac{1}{2}\sqrt{\gamma\eta}.$$

Let $t_1' = \frac{\sqrt{C\gamma/\eta}}{\frac{1}{2}\sqrt{\gamma\eta}} = 2\sqrt{C}\eta^{-1}$, then $t_1' - 1 < t_0$ and thus $x_{t_1'} < \frac{1}{2}\sqrt{\gamma\eta}$, i.e., $t_1 < t_0$ must holds. $\qquad\square$

## G.3 Final Phase: $y_t$ Gets Trapped Above the Origin

Now, we are going to consider the final-stage behavior of USAM.

**Theorem 44** (Main Conclusion of Final Phase; USAM Case). *After Initial Phase, we always have $|x_t| \leq \sqrt{C\gamma\eta}$ and $|y_t| \leq \sqrt{C\gamma/\eta}$. Moreover, once we have $\eta(1 + \rho y_t^2)y_t^2 = 2 - \epsilon$ for some $\mathfrak{t} \geq t_1$ and $\epsilon > 0$, we must have $|x_{t+1}| \leq \exp(-\Omega(\epsilon))|x_t|$ for all $t \geq \mathfrak{t}$. This consequently infers that $y_\infty^2$, which is defined as $\liminf_{t\to\infty} y_t^2$, satisfies $y_\infty^2 \geq (1 - 4C\gamma(\eta + \rho C\gamma)^2\epsilon^{-1})y_t^2$. As $\epsilon$ is a constant independent of $\eta$ and $\rho$ and can be arbitrarily close to 0, this shows the second claim of Theorem 35.*

*Proof.* The first conclusion is an analog of Lemma 28 and Lemma 27 (which allows a simpler analysis thanks to the removal of normalization), as we will show in Lemma 45. The second part requires a similar (but much more sophisticated) analysis to Lemma 15 of Ahn et al. (2023a), which we will cover in Lemma 47. □

**Lemma 45.** *Suppose that $|x_t| \leq \sqrt{C\gamma\eta}$, $|y_t| \leq \sqrt{C\gamma/\eta}$. Assuming $\eta\rho \leq 1$, then $|x_{t+1}| \leq \sqrt{C\gamma\eta}$ and $|y_{t+1}| \leq \sqrt{C\gamma/\eta}$ as well. Furthermore, $|x_t| \leq \sqrt{C\gamma\eta}$ and $|y_t| \leq \sqrt{C\gamma/\eta}$ hold for all $t > t_1$.*

*Proof.* Suppose that $x_t \geq 0$ without loss of generality. The case where $x_{t+1} \geq 0$ is trivial by Lemma 37. Otherwise, using Lemma 36 and Equation 20, we can write

$$x_{t+1} = x_t - \eta\ell_t' \cdot (y_t + \rho\tilde{\ell}_t' x_t) = x_t - \eta|\ell_t'| \cdot (|y_t| + \rho|\tilde{\ell}_t'|x_t).$$

By Assumption (A1), $|\ell_t'|$ and $|\tilde{\ell}_t'|$ are bounded by 1. By the condition that $\eta\rho \leq 1$,

$$x_{t+1} \geq (1 - \eta\rho)x_t - \eta|y_t| \geq -\sqrt{C\gamma\eta},$$

where we used $|y_t| \leq sqrt{C\gamma/\eta}$. Similarly, for $y_{t+1}$, only considering the case where $y_t \geq 0$ and $y_{t+1} \leq 0$ suffices. We have the following by symmetry

$$y_{t+1} = y_t - \eta\ell_t' \cdot (x_t + \rho\tilde{\ell}_t' y_t) \geq y_t - \eta(|x_t| + \rho y_t) = (1 - \eta\rho)y_t - \eta|x_t| \geq -\eta^{1.5}\sqrt{C\gamma} \geq -\sqrt{C\gamma/\eta},$$

where we used $|x_t| \leq \sqrt{C\gamma\eta}$.

The second part of the conclusion is done by induction. According to the definition of Initial Phase, we have $|x_{t_1+1}| \leq \frac{1}{2}\sqrt{\gamma\eta}$. As $C \geq 1 \geq \frac{1}{4}$, we consequently have $|x_t| \leq \sqrt{C\gamma\eta}$ for all $t > t_1$ from the first part of the conclusion. Regarding $|y_t|$, recall Lemma 40 infers $y_t > 0$ for all $t \leq t_1 + 1$ and Lemma 37 infers the monotonicity of $y_t$, we have $y_{t_1+1} \leq y_0 = \sqrt{C\gamma/\eta}$. Hence, $|y_t| \leq \sqrt{C\gamma/\eta}$ for all $t > t_1$ as well. □

Before showing the ultimate conclusion Lemma 47, we first show the following single-step lemma:

**Lemma 46.** *Suppose that $\eta(1 + \rho y_t^2)y_t^2 < 2$ and $|x_t| \leq \sqrt{C\gamma\eta}$ for some t. Define $\epsilon_t = 2 - \eta(1 + \rho y_t^2)y_t^2$ (then we must have $\epsilon_t \in (0, 2)$). Then we have*

$$|x_{t+1}| \leq |x_t| \exp\left(-\min\left\{(1 + \rho C\gamma\eta)\epsilon_t, \frac{2 - \epsilon_t}{8}\right\}\right). \tag{22}$$

*Proof.* Without loss of generality, assume that $x_t > 0$. From Equation 20, we can write:

$$x_{t+1} = x_t - \eta\ell_t' \cdot (y_t + \rho\tilde{\ell}_t' x_t) = x_t - \eta|\ell_t'| \cdot (|y_t| + \rho|\tilde{\ell}_t'|x_t),$$

where we used $\text{sign}(\ell_t') = \text{sign}(\tilde{\ell}_t') = \text{sign}(x_t y_t) = \text{sign}(y_t)$ (Lemma 36). By Assumption (A2),

$$\begin{aligned}
x_{t+1} &\geq x_t - \eta\left|(x_t + \rho\tilde{\ell}_t' y_t)(y_t + \rho\tilde{\ell}_t' x_t)\right|(|y_t| + \rho|x_t y_t|x_t) \\
&= x_t - \eta(x_t + \rho|\tilde{\ell}_t'||y_t|)(|y_t| + \rho|\tilde{\ell}_t'|x_t)(|y_t| + \rho x_t^2|y_t|) \\
&\geq x_t - \eta(x_t + \rho x_t y_t^2)(|y_t| + \rho x_t^2|y_t|)(|y_t| + \rho x_t^2|y_t|) \\
&= (1 - \eta(1 + \rho y_t^2)(1 + \rho x_t^2)^2 y_t^2)x_t \\
&\geq (1 - (1 + \rho C\gamma\eta)\eta(1 + \rho y_t^2)y_t^2)x_t \\
&\geq -(1 - (1 + \rho C\gamma\eta)\epsilon_t)x_t, \tag{23}
\end{aligned}$$

where we used $|x_t| \leq \sqrt{C\gamma\eta}$. For the other direction, we have the following by Assumption (A3):

$$
\begin{aligned}
x_{t+1} &\leq x_t - \eta \frac{|(x_t + \rho\tilde{\ell}'_t y_t)(y_t + \rho\tilde{\ell}'_t x_t)|}{2} \left( |y_t| + \rho \frac{|x_t y_t|}{2} x_t \right) \\
&= x_t - \eta \frac{(x_t + \rho|\tilde{\ell}'_t||y_t|)(|y_t| + \rho|\tilde{\ell}'_t|x_t)}{2} \left( |y_t| + \rho \frac{x_t^2}{2}|y_t| \right) \\
&\leq x_t - \eta \frac{(x_t + \rho x_t y_t^2)(|y_t| + \rho|y_t|x_t^2)}{8} \left( |y_t| + \rho \frac{x_t^2}{2}|y_t| \right) \\
&= \left( 1 - \frac{\eta}{8}(1 + \rho y_t^2)(1 + \rho x_t^2)\left( 1 + \rho \frac{x_t^2}{2} \right)y_t^2 \right) x_t \\
&\leq \left( 1 - \frac{\eta}{8}(1 + \rho y_t^2)y_t^2 \right) x_t = \left( 1 - \frac{2 - \epsilon}{8} \right) x_t, \quad (24)
\end{aligned}
$$

where we used $(1 + \rho x_t^2) \geq 1$ and $(1 + \rho \frac{x_t^2}{2}) \geq 1$. Equation 22 follows from Equation 23 and Equation 24. □

**Lemma 47.** *Let* $\mathfrak{t}$ *be such that i)* $\eta(1 + \rho y_{\mathfrak{t}}^2)y_{\mathfrak{t}}^2 = 2 - \epsilon$ *where* $\epsilon \in (0, \frac{2}{9})$ *is a constant, and ii)* $|x_{\mathfrak{t}}| \leq \sqrt{C\gamma\eta}$. *Then we have the following conclusion on* $\liminf_{t \to \infty} y_t^2$, *denoted by* $y_\infty^2$ *in short:*

$$
y_\infty^2 = \liminf_{t \to \infty} y_t^2 \geq \left( 1 - 4C\gamma(\eta + \rho C\gamma)^2\epsilon^{-1} \right)y_{\mathfrak{t}}^2.
$$

While the $\epsilon^{-1}$ looks enormous, it is a constant independent of $\eta$ and $\rho$; in other words, we are allowed to set $\epsilon$ as close to zero as we want. As we only consider the dependency on $\eta$ and $\rho$, we can abbreviate this conclusion as $y_\infty^2 \geq (1 - \mathcal{O}(\eta^2 + \rho^2))y_{\mathfrak{t}}^2$, as we claim in the main text.

*Proof.* In analog to Equation 23, we derive the following for $y_t$:

$$
y_{t+1}^2 \geq \left( 1 - \eta\left(1 + \rho y_t^2\right)^2(1 + \rho x_t^2)x_t^2 \right)^2 y_t^2 \geq \left( 1 - 2\eta(1 + \rho C\gamma/\eta)^2(1 + \rho C\gamma\eta)x_t^2 \right)y_t^2,
$$

where the second inequality uses Lemma 45. Let $d_t = y_{\mathfrak{t}}^2 - y_t^2$, then we have

$$
d_{t+1} \leq d_t + 2\eta(1 + \rho C\gamma/\eta)^2(1 + \rho C\gamma\eta)x_t^2 y_t^2 \leq d_t + 4\eta^{-1}(\eta + \rho C\gamma)^2 y_t^2 x_t^2,
$$

where we used the assumption that $\rho C\gamma\eta \leq 1$ and the fact that $y_t^2$ is monotonic (thus $y_t^2 \leq y_{\mathfrak{t}}^2$).

According to Equation 22, we have $|x_{t+1}| \leq |x_t| \exp(-\Omega(\epsilon))$ for all $t \geq t_2$. Hence, we have

$$
d_\infty = \limsup_{t \to \infty} d_t \leq d_{\mathfrak{t}} + \sum_{t=\mathfrak{t}}^\infty 4\eta^{-1}(\eta + \rho C\gamma)^2 y_{\mathfrak{t}}^2 x_t^2 = 4\eta^{-1}(\eta + \rho C\gamma)^2 y_{\mathfrak{t}}^2 \cdot \epsilon^{-1} x_{\mathfrak{t}}^2,
$$

where we used $d_{\mathfrak{t}} = 0$ (by definition) and the sum of geometric series. Plugging back $x_{\mathfrak{t}}^2 \leq C\gamma\eta$,

$$
y_\infty^2 = \liminf_{t \to \infty} y_t^2 \geq y_{\mathfrak{t}}^2 - 4C\gamma(\eta + \rho C\gamma)^2 y_{\mathfrak{t}}^2,
$$

as claimed. □

# H  Omitted Proof of USAM Over General PL functions

**Theorem 48** (Formal Version of Theorem 6). *For any* $\mu$-*PL and* $\beta$-*smooth loss function* $\mathcal{L}$, *for any learning rate* $\eta < 1/\beta$ *and* $\rho < 1/\beta$, *for any initialization* $w_0$, *the following holds for USAM:*

$$
\|w_t - w_0\| \leq \eta(1 + \beta\rho)\sqrt{\frac{2\beta^2}{\mu}}\left( 1 - 2\mu\eta(1 - \rho\beta)\left( \frac{\eta\beta}{2}(1 - \rho\beta) \right) \right)^{-1/2}\sqrt{\mathcal{L}(w_0) - \mathcal{L}^*}, \quad \forall t \geq 0,
$$

*where* $\mathcal{L}^*$ *is the short-hand notation for* $\min_w \mathcal{L}(w)$.

We first state the following useful result by Andriushchenko and Flammarion (2022, Theorem 10).

**Lemma 49** (Descent Lemma of USAM over Smooth and PL Losses). *For any $\beta$-smooth and $\mu$-PL loss function $\mathcal{L}$, for any learning rate $\eta < 1/\beta$ and $\rho < 1/\beta$, the following holds for USAM:*

$$\mathcal{L}(w_t) - \mathcal{L}^\star \leq \left(1 - 2\mu\eta(1-\rho\beta)\left(1 - \frac{\eta\beta}{2}(1-\rho\beta)\right)\right)^t (\mathcal{L}(w_0) - \mathcal{L}^\star), \quad \forall t \geq 0.$$

*Proof of Theorem 48.* We follow the convention in Karimi et al. (2016): Let $\mathcal{X}^\star$ be the set of global minima and $x_p$ be the projection of $x$ onto the solution set $\mathcal{X}^\star$. From $\beta$-smoothness, it follows that

$$\|\nabla\mathcal{L}(x)\| = \|\nabla\mathcal{L}(x) - \nabla\mathcal{L}(x_p)\| \leq \beta\|x - x_p\|, \quad \forall x.$$

Now since $\mathcal{L}$ is $\beta$-smooth and $\mu$-PL, Theorem 2 from (Karimi et al., 2016) implies that the quadratic growth condition holds, i.e.,

$$\frac{2}{\mu}(\mathcal{L}(x) - \mathcal{L}^\star) \geq \|x - x_p\|^2, \quad \forall x.$$

Thus, it follows that

$$\|\nabla\mathcal{L}(x)\|^2 \leq \frac{2\beta^2}{\mu}(\mathcal{L}(x) - \mathcal{L}^\star), \quad \forall x.$$

Moreover, from $\beta$-smoothness, we have

$$\|\nabla\mathcal{L}(x + \rho\nabla\mathcal{L}(x))\| \leq \|\nabla\mathcal{L}(x)\| + \beta\|\rho\nabla\mathcal{L}(x)\| = (1 + \beta\rho)\|\nabla\mathcal{L}(x)\|, \quad \forall x.$$

Thus, by the update rule of USAM (2), it follows that

$$\|w_t - w_0\| \leq \eta\sum_{i=0}^{t-1}\|\nabla\mathcal{L}(w_i + \rho\nabla\mathcal{L}(w_i))\|$$

$$\leq \eta(1 + \beta\rho)\sum_{i=0}^{t-1}\|\nabla\mathcal{L}(w_i)\|$$

$$\leq \eta(1 + \beta\rho)\sum_{i=0}^{t-1}\sqrt{\frac{2\beta^2}{\mu}(\mathcal{L}(w_i) - \mathcal{L}^\star)}$$

$$= \eta(1 + \beta\rho)\sqrt{\frac{2\beta^2}{\mu}}\sum_{i=0}^{t-1}\sqrt{\mathcal{L}(w_i) - \mathcal{L}^\star}.$$

Now we only to invoke the USAM descent lemma stated before, i.e., Lemma 49, giving

$$\sum_{i=0}^{t-1}\sqrt{\mathcal{L}(w_i) - \mathcal{L}^*} \leq \sum_{i=0}^{t-1}\left(1 - 2\mu\eta(1-\rho\beta)\left(1 - \frac{\eta\beta}{2}(1-\rho\beta)\right)\right)^{i/2}\sqrt{\mathcal{L}(w_0) - \mathcal{L}^*}$$

$$\leq \left(1 - 2\mu\eta(1-\rho\beta)\left(\frac{\eta\beta}{2}(1-\rho\beta)\right)\right)^{-1/2}\sqrt{\mathcal{L}(w_0) - \mathcal{L}^*}.$$

Putting the last two inequalities together then give our conclusion. □

