# OpenReview forum: "The Crucial Role of Normalization in Sharpness-Aware Minimization"
_NeurIPS.cc/2023/Conference — NeurIPS 2023 poster_

### Official Review · Reviewer_a57m · 2023-06-26

**Soundness:** 3 good
**Presentation:** 3 good
**Contribution:** 2 fair
**Rating:** 5
**Confidence:** 3

**Summary:**

This work investigates the role of the normalization factor $1/||\nabla L(x,w)||$ in the perturbation step of the Sharpness-Aware Minimization (SAM) algorithm. To this end, the authors compare the behavior of gradient descent, SAM and a version of SAM without the normalization (USAM) via experiments and extensive theoretical analysis. The authors find that USAM lacks some of SAMs beneficial properties: On the one hand, it is harder to choose an appropriate perturbation radius for USAM and on the other hand, the ‘drifting along minima’, which is thought to be connected to SAMs empirical success, only occurs in a restricted manner.

**Strengths:**

The paper is very well written and polished. The motivation (understanding the effects of the normalization factor in the perturbation step of SAM) is outlined clearly. The theoretical analysis is thorough. The toy experiments are clever and well-designed to isolate the effects that are to be investigated, and mostly support and illustrate the conclusions arising from the theoretical analysis.

**Weaknesses:**

The authors essentially provide extensive theoretical analysis and toy experiments on why USAM works worse than SAM, which often stems from the fact that the gradient magnitude changes during training. USAM is however not a relevant algorithm in practice - it was only introduced as a simplified version of SAM for easier theoretical analysis. The insights on why SAM works in the first place are thus limited (e.g. that SAM can drift along minima was analyzed in [1], and adding that normalization is important for this to happen adds little to the understanding of SAM).

Apart from [2], where USAM was found to work similarly to SAM for a ResNet on CIFAR, it is further unclear how well USAM works in realistic scenarios. While the authors provide some insights (in 2.3 and 3.3), the settings in those experiments are fairly restrictive (one focuses on early-stage training and the other on an adversarial initialisation) and (nicely) highlight isolated effects when dropping the normalization factor. While I think that negative results (like, _In contrast to SAM, USAM does not work in practice, and here we explain why_) can be of value to the community, it is thus unclear to me if the results presented in the paper are indeed relevant for practical neural network training with SAM (or USAM). Presenting results of several models trained with SGD, SAM and USAM on e.g. CIFAR10 & CIFAR100 over a range of $\rho$ values would bring insights on this and bridge the gap between the analysis and potential practical implications of this paper. In particular, one could e.g. show that for practical deep learning settings picking $\rho$ is indeed more difficult, or that USAM does not find well-generalizing minima (due to the restricted drifting along minima).

[1] Peter L. Bartlett, Philip M. Long and Olivier Bousque. The Dynamics of Sharpness-Aware Minimization: Bouncing Across Ravines and Drifting Towards Wide Minima. Arxiv preprint

[2] Maksym Andriushchenko and Nicolas Flammarion. Towards understanding sharpness-aware Minimization. ICML 2022

**Questions:**

As outlined above, my main question is whether the results and claims regarding USAM can indeed be observed in realistic settings for neural network training. This could e.g. be addressed by training a range of models on CIFAR100 for a range of $\rho$ values and plotting the performance of each algorithm in dependence of $\rho$ (like e.g. in Figure 4 of [1], just with one such plot per model).

Minor questions:
- Figure 7: How was $\rho$ chosen? Is it possible that the optimal $\rho$ is different for SAM and USAM, i.e. that for a ‘good’ choice of $\rho$ USAM would also behave like SAM for the selected range of $\eta$?
- Section 2.3: Could avoiding the SAM-perturbation at the early stages of training mitigate the problem of divergence for USAM?
- Figure 8 in Appendix B: I cannot see the purple line for which USAM diverges ($\rho=0.5$). Is it not shown in the plot?
- Appendix D.2: ‘USAM diverges with small $\rho$’ - shouldn’t this be ‘large $\rho$’?

[1] Jungmin Kwon, Jeongseop Kim, Hyunseo Park, and In Kwon Choi. Asam: Adaptive sharpness-aware
minimization for scale-invariant learning of deep neural networks. ICML 2021

**Limitations:**

I don’t see potential negative societal impact that the authors should have addressed.

---

> ### Author Rebuttal · Authors · 2023-08-09
>
> Thank you for your insightful comments! Here are our responses:
>
> ---
> ### Why Compare SAM to USAM?
>
> We agree that USAM is by no means practical. However, the reason for comparison is solely to understand SAM better and doesn't have to do with USAM's impracticality. In particular, we are trying to **understand the success of SAM in practice rigorously and theoretically**: It is undoubtedly that SAM exhibits a ground-breaking success in practice (getting ~700 citations in ~2 years); thus, better understanding the success of SAM is critical for designing better algorithms in the future.
>
> Indeed, extensive effort has been made to answer this question in the theoretical ML literature. However, we notice that **most of such works assume that the normalization of SAM can be omitted "for simplicity"** (e.g., Andriushchenko and Flammarion (2022); Behdin and Mazumder (2023); Agarwala and Dauphin (2023); Kim et al. (2023)). However, as far as we know, **the only rationale behind such a simplification is just one experiment** on CIFAR10 conducted by Andriushchenko and Flammarion (2022).
>
> In our paper, we take a step back and **challenge this common "simplification", asking whether normalization plays a critical role in SAM**. To make our results scientifically rigorous, our paper considers the variant of SAM by **only** removing the normalization, namely USAM. This provides a precise and scientific ablation study. Hence, **we did not consider USAM for its practicality** but **for providing a precise and scientific understanding of SAM**. By considering USAM, we successfully and rigorously identify (one of) the reasons behind SAM's success, namely the normalization step.
>
> Finally, we would like to remark that **previous papers do not yet achieve such an accurate identification**. For example, as you mentioned, Wen et al. (2023) show that SAM can drift along the minima manifold for a "flatter" minimum. However, the crucial role played by the normalization step is beyond the scope of their paper. In our work, by comparing USAM and SAM, we identify its crucial role.
>
> ---
> ### More Experiments on Larger Datasets
>
> Thanks for mentioning this. It would be great to conduct large-scale experiments on larger datasets like CIFAR-100 and larger neural networks, as it can further verify our theoretical conclusions. Although we won't be able to share the full results this time, we will attempt to add some experimental results in our final version.
>
> We also would like to highlight that **our paper is a theoretical paper aiming at identifying the reason behind SAM's success**; and we did succeed in justifying the crucial role of normalization by *i)* rigorously comparing SAM's behavior with the un-normalized variant USAM in theory, and *ii)* carefully designing experiments inspired by our theory to support our claim via real-world models (ResNet18 on CIFAR-10).
>
> ---
> ### Should the Choice of $\rho$ Be Different between SAM and USAM?
>
> Yes, the optimal tuning of algorithmic parameters $(\eta,\rho)$ likely differs between SAM and USAM. However, we emphasize that **we are not comparing the performance of SAM and USAM** when fixing the same $(\eta,\rho)$. Instead, **we are comparing the robustness of SAM and USAM** to parameter tuning (i.e., $\eta$ and $\rho$).
>
> In other words, our experiments like Figures 7 and 8 are **not** for "fixing a same $(\rho,\eta)$ and comparing the behavior of SAM and USAM"; instead, we consider "**how much does change $\eta$ or $\rho$ affects their algorithmic behaviors**". Specifically,
> 1. In Figure 7, we fix $\rho$ and use various different $\eta$'s. We discover that fixing the same $\rho$, SAM with different $\eta$'s finds the same optimum; however, USAM with different $\eta$ behaves pretty differently. Thus, **Figure 7 illustrates that USAM is less robust to the tuning of $\eta$**.
> 2. In Figure 8, we instead fix $\eta$ and try different $\rho$'s. It turns out that SAM is still robust to different $\rho$'s while USAM is not. Again, we are not comparing the performance of SAM and USAM with the same $(\eta,\rho)$; instead, we identify the effect of varying $\rho$ on the algorithmic behaviors. **Our conclusion from Figure 7 generalizes to $\rho$ as well**.
>
> Hence, putting Figures 7 and 8 together, we conclude that **SAM is robust to different configurations of $(\eta,\rho)$ while USAM is robust to neither of them**. We will make our conclusion more transparent in the final version.
>
> ---
> ### Does Removing Perturbation in the Early Phase of USAM Resolves Divergence?
>
> We believe it is possible as removing normalization in the early stage essentially makes USAM same as GD. This is also our motivation for Section 3, where we illustrate that **"USAM without perturbation in its early phase" cannot guarantee a good performance, either**.
>
> As our Figure 5 suggests, USAM loses the ability to find a better global minimum, matching our conclusions in Theorems 5 and 6. This indicates that **normalization is not only important in the initial stages, but also critical in the final phase in order to escape bad minima**. Consequently, removing perturbation in the early phase of USAM only resolves the divergence issue but still cannot ensure better performance.
>
>  ----
> ### USAM Diverges with "Small" or "Large" $\rho$?
>
> Thanks for mentioning this! We indeed meant "small $\rho$". According to Theorem 15, USAM diverges when $\rho\approx 15\eta=o(1)$; however, by Theorem 12, SAM doesn't diverge even when $\rho=\Omega(1)$. Hence, **USAM starts to diverge with a "small" $\rho$ compared to SAM**. We will change the titles of Appendices D.1 and D.2 to "SAM Converges with a Large $\rho$" and "USAM Diverges from a Small $\rho$", respectively.
>
> ---
> Thank you once again for taking the time to review our paper. Given that you have not raised any concerns regarding the correctness or novelty of our work, if you believe that we have addressed your points, we would appreciate it if you would consider raising your score.

---

> > ### Comment · Reviewer_a57m · 2023-08-11
> >
> > Thanks for your response.
> >
> > - Regarding my question w.r.t. Figure 7:
> > I understand that you fix $\rho$ and vary $\eta$ (and vice versa in Figure 8), but my question was whether the plot might look different if another $\rho$ that might be more favorable for USAM were chosen, e.g. $\rho=0.3$ seems to work somewhat well for USAM in Figure 8. In my understanding, in its current form _”Figure 7 illustrates that USAM is less robust to the tuning of_ $\eta$” __for this particular choice__ of $\rho$. Overall, it might make sense to report a full grid-search over ($\eta$, $\rho$).
> >
> > - Regarding my main question:
> > I understand that this work is of theoretical nature, and that USAM is only analyzed in order to investigate the role of normalization in SAM. I further agree that SAM exhibits success in practice, and that a better understanding of the success of SAM is important.
> > Thus, “challenging this common simplification, asking whether normalization plays a critical role in SAM” can be a valuable contribution, as I also outlined in the main review.
> > As also outlined before, I am however not convinced that USAM is indeed _“impractical”_. In particular, I see very little (empirical) evidence that _“the success of SAM is (at least in part) backed up by the normalization”_ for practical settings. In contrast, there is some evidence that USAM might work just as well (Andriushchenko 2022). In this, I agree with with
> > 	1. reviewer chBL, asking for _“insights into the differences between SAM and USAM during real training from scratch”_,
> > 	2. reviewer FoGd (_“Empirical verification on real-world tasks is limited.”_)
> > 	3. reviewer pAM7 ( _“Additionally, previous theoretical works have indicated that USAM can be tuned to achieve similar performance to SAM (Andriushchenko 2022). Though there exists a difference between USAM and SAM in the toy models in the paper, there aren't any experiments that indicate that not normalizing is a problem on real datasets and models (that USAM requires delicate tuning, or doesn't work at all in terms of achieve similar training curve and test performance as SAM). “_)
> >
> > 	This is also the core of my criticism - I think it is unclear if normalization is actually crucial in realistic settings, but this is substantial in order to make the (well conducted) investigations of the paper relevant, even though it is a theory paper. If USAM were to work just as well as SAM in practical settings, the conclusions from this work would be totally different.

---

> > > ### Author Response · Authors · 2023-08-21
> > >
> > > Thanks for your follow-up comment! We agree that extending Figures 7 and 8 with more configurations of $(\eta,\rho)$ can be beneficial. As we cannot add links or attachments in this phase, we will try to add them in our final version.
> > >
> > > Regarding your main question, thank you for finding our theoretical contribution meaningful and valuable! By "impractical", we meant that USAM is never utilized in practice (i.e., it is never adopted by works of empirical nature). We agree that Andriushchenko et al. (2022) provide evidence that USAM may work; however, we insist that our experiments on ResNet18 over CIFAR-10 are also "practical settings" where we illustrate that USAM may not work.
> > >
> > > Given these two contradictory results on where USAM behaves worse than SAM in practice, it is valuable to prove or disprove this! If we manage to secure access to additional computational resources, we would happily add more experiments. We would also mention this in our final revision as an open problem.
> > >
> > > We thank you again for your time and efforts in reviewing our paper. We benefit greatly from your comments and will try our best to conduct more experiments that can support our claims. We agree that resolving the inconsistency between our results and that of Andriushchenko et al. (2022) is crucial -- however, this does not imply that either Andriushchenko et al. (2022) or our results are “wrong”. Instead, it raises the significance of the role of normalization, and we believe it is essential to surface these findings to the broader community.

---

> > > > ### Comment · Reviewer_a57m · 2023-08-21
> > > > **Summary**
> > > >
> > > > After considering the rebuttal and discussion period, my perspective remains largely the same:
> > > > I still think the analysis in this work is well conducted (yet somewhat toyish in places) and could potentially bring insights into the underlying mechanisms of SAM. However, the conclusions that can be drawn are unclear, and this could be easily cleared up when backed with more empirical analysis. It would be much more convincing to start from observations and then provide the present analysis as explanation, instead of providing explanations for why _“the success of SAM is (at least in part) backed up by the normalization”_, when this is not entirely clear. The most relevant conclusion I can see now is to better be safe than sorry and take normalization into account for theoretical analysis, even though in practice we don’t really know if it matters or not.
> > > >
> > > > I have updated my score since I do not see bigger technical issues with the analysis, but would really like to urge the authors to include the suggested experiments and adjust potential conclusions accordingly in a revised version.

---

### Official Review · Reviewer_FoGd · 2023-07-04

**Soundness:** 3 good
**Presentation:** 3 good
**Contribution:** 2 fair
**Rating:** 5
**Confidence:** 2

**Summary:**

The paper investigates the role of the normalization term in the recently proposed optimization algorithm Sharpness-Aware Minimization (SAM). The authors theoretically, and empirically, study the differences among  SAM, its un-normalized counterpart USAM, and gradient descent, under multiple settings (strongly convex and smooth losses, a non-convex case, single-neuron linear networks, sparse coding), and conclude with two key takeaways: normalization in SAM helps (1) stabilize algorithm iterates, and (2) enables the algorithm to drift near minima.

**Strengths:**

- The paper thoroughly investigates the role of normalization across many different settings, and convincingly show the benefit of the normalization term.
- The authors provide clear key takeaways for why normalization helps, with both empirical and theoretical support.
- The paper makes a solid contribution to our understanding of why normalization is beneficial
- The authors compared the impact of tuning the learning rate and perturbation radius between SAM and USAM

**Weaknesses:**

- Empirical verification on real-world tasks is limited. For example, generating more results similar to Figure 5 would strengthen the papers conclusions.
- While having a deeper understanding of a specific term in an optimizer adds value, the overall question of the work does not feel like a pressing issue in ML. The takeaway of the rigorous work is to use the original version of SAM, which most people who use SAM already do.

**Questions:**

- Can the takeaways from this work be generalized to the benefits of other optimizers that use normalization (such as normalized gradient descent)?
- Are there any advantages for USAM over SAM?

**Limitations:**

No, the authors do not discuss limitations and potential negative societal impact.

---

> ### Author Rebuttal · Authors · 2023-08-09
>
> Thank you for your insightful comments! Here are our responses:
>
> ---
>
> ### Is the Main Takeaway "Using SAM instead of USAM"?
>
> The takeaway is not this one if interpreted literally, but slightly more nuanced. Let us summarize the **two main takeaways** of our paper, one for the theoretical ML community and one for the empirical ML community:
>
> 1. For the theoretical ML community, our main takeaway is **normalization plays a critical role and cannot be neglected when analyzing SAM** -- when investigating the success of SAM (which is important for future algorithm designs), normalization plays a non-negligible role; removing normalization can drastically change the algorithmic behaviors.
>
> 2. For the empirical ML community, the main takeaway is **the practicality of SAM is (in part) backed up by the normalization**. Previously, empirical researchers successfully applied SAM in various applications, and theoretical researchers characterized SAM's ability to perform well. However, the reason behind such a success is still unclear until our work. Our work suggests that **when designing other algorithms in the future, whether using normalization should be taken seriously**.
>
> ---
>
> ### Advantages of USAM over SAM?
>
> Thanks for mentioning this. The only advantage of USAM over SAM that we are aware of is its simplicity in analyses -- and this is probably the main reason why various previous theoretical works focused on USAM instead of SAM (e.g., Andriushchenko and Flammarion (2022); Behdin and Mazumder (2023); Agarwala and Dauphin (2023); Kim et al. (2023)).
>
> Still, we would like to mention that **the main reason for considering USAM is not its practicability or any other advantage** (more details can be found in our first response to Reviewer a57m or Reviewer pAM7). For context, we aim to understand the considerable success of SAM in practice. However, we noted that existing theoretical works mostly ignore the normalization in SAM "for simplicity". Our work challenges this "simplification" and **justifies that normalization plays a critical role**. We only remove the normalization in SAM and keep all other parts unchanged for a scientifically rigorous conclusion. This gives USAM -- hence, **the reason for considering USAM is for a precise ablation study**, not its practicability (or any other advantage).
>
>
> > ### References
> > * Maksym Andriushchenko and Nicolas Flammarion. Towards Understanding Sharpness-Aware Minimization. *Proceedings of the 39th International Conference on Machine Learning*, PMLR 162:639-668, 2022.
> > * Kayhan Behdin and Rahul Mazumder. On Statistical Properties of Sharpness-Aware Minimization: Provable Guarantees. *arXiv preprint arXiv:2302.11836*, 2023.
> > * Atish Agarwala, Yann Dauphin. SAM operates far from home: eigenvalue regularization as a dynamical phenomenon. *Proceedings of the 40th International Conference on Machine Learning*, PMLR 202:152-168, 2023.
> > * Hoki Kim, Jinseong Park, Yujin Choi, and Jaewook Lee. Stability Analysis of Sharpness-Aware Minimization. *arXiv preprint arXiv:2301.06308*, 2023.
>
> ---
>
> ### More Experiments
>
> Thanks for mentioning this! We conducted another experiment similar to Figure 5, where we used a more realistic "full-batch checkpoint" instead of an "adversarial checkpoint" (see our second response to Reviewer chBL for more details); indeed, our conclusion still holds. The plot is contained in our attachment.
>
> It would definitely be great to conduct large-scale experiments on larger datasets like CIFAR-100 and larger neural networks, as it can further verify our theoretical conclusions. Although we won't be able to share the full results this time, we will make attempts to add some experimental results in our final version.
>
> We also would like to highlight that **our paper is a theoretical paper aiming at identifying the reason behind SAM's success**; and, we did succeed in justifying the crucial role of normalization by *i)* rigorously comparing SAM's behavior with the un-normalized variant USAM in theory, and *ii)* carefully designing experiments inspired by our theory to support our claim via real-world models (ResNet18 on CIFAR-10).
>
> ---
>
> Thank you once again for taking the time to review our paper. Given that you have not raised any concerns regarding the correctness or novelty of our work, if you believe that we have addressed your points, we would appreciate it if you would consider raising your score.

---

> > ### Comment · Reviewer_FoGd · 2023-08-15
> >
> > Thank you for the rebuttal and clarifications. My mention of limited real-world experiments does not imply the need for large-scale experiments.
> >
> > I decided to keep my score at borderline accept.

---

### Official Review · Reviewer_chBL · 2023-07-06

**Soundness:** 3 good
**Presentation:** 4 excellent
**Contribution:** 3 good
**Rating:** 6
**Confidence:** 4

**Summary:**

- The paper investigates the difference between SAM and Unclipped SAM (USAM) in terms of stabilizing and drifting along the manifold.
- The paper reveals a significant disadvantage in USAM compared to SAM, which was previously thought to be similar in various papers.
- The paper provides mathematical and experimental evidence to support their ideas in diverse settings.

**Strengths:**

- Overall, the idea of investigating the difference between SAM and USAM is clear and interesting.
- The paper identifies the role of normalization and highlights that USAM behaves more like GD than SAM, potentially getting stuck in local minima and requiring careful tuning.
- The mathematical proofs are clear and offer unique insights compared to previous papers, such as [1].

[1] An SDE for Modeling SAM: Theory and Insights (ICML’23)

**Weaknesses:**

Please refer to questions.

**Questions:**

- I believe the selection of the radius $\rho$ in USAM and SAM should be different. While I understand that the authors aimed to test various values of $\rho$, SAM and USAM may require different optimal radius settings to achieve similar results as mentioned in [2]. The chosen radius sets seem to be selected naively without obtaining the best results in each setting and needs to be improved.
- [2] proposed that SAM and USAM perform similarly in real training problems. However, your experimental results rely on special settings with "adversarial checkpoints." Can you provide insights into the differences between SAM and USAM during "real training" from scratch, rather than in the adversarial context?
- Instead of scheduling based on normalization, have you considered linear scheduling with respect to the learning rate or other scheduling approaches similar to learning rate selection? Additionally, if the early phase of training is crucial, normalizing the radius first and then using a fixed $\rho$ for that phase may be suitable for your experiments. Could you provide additional experimental results using scheduling methods other than normalization?
- [3] mentioned that the early stage of training, referred to as the "Edge of stability," differs between SAM and SGD rather than focusing on the property of SAM to be beneficial in later training stages [2]. However, their analysis was conducted using USAM. If USAM is not suitable for sharpness-aware training, how do you interpret their results in relation to your ideas, and how do they compare to your findings?

I’d be happy to raise my score if the authors can address the weaknesses/questions/limitations in the rebuttal.


[2] Towards Understanding Sharpness-Aware Minimization (ICML’22)

[3] SAM operates far from home: eigenvalue regularization as a dynamical phenomenon (ICML’23)



**Limitations:**

The paper has not provided a detailed discussion on limitations.

---

> ### Author Rebuttal · Authors · 2023-08-09
>
> Thank you for your insightful comments! Here are our responses:
>
> ---
>
> ### Should the Choice of $\rho$ Be Different in SAM and USAM?
>
> Yes, the optimal tuning of algorithmic parameters $(\eta,\rho)$ likely differs between SAM and USAM. However, we would like to emphasize that **we are not comparing the performance of SAM and USAM** when fixing the same $(\eta,\rho)$. Instead, **we are comparing the robustness of SAM and USAM** to parameter tuning (i.e., $\eta$ and $\rho$).
>
> In other words, our experiments like Figures 7 and 8 are **not** for "fixing a same $(\rho,\eta)$ and comparing the behavior of SAM and USAM"; instead, we consider "**how much does change $\eta$ or $\rho$ affects their algorithmic behaviors**". Specifically,
> 1. In Figure 7, we fix $\rho$ and use various different $\eta$'s. We discover that fixing the same $\rho$, SAM with different $\eta$'s finds the same optimum; however, USAM with different $\eta$ behaves pretty differently. Thus, **Figure 7 illustrates that USAM is less robust to the tuning of $\eta$**.
>
> 3. In Figure 8, we instead fix $\eta$ and try different $\rho$'s. It turns out that SAM is still robust to different $\rho$'s while USAM is not. Again, we are not comparing the performance of SAM and USAM with the same $(\eta,\rho)$; instead, we identify the effect of varying $\rho$ on the algorithmic behaviors. **Our conclusion from Figure 7 generalizes to $\rho$ as well**.
>
> Hence, putting Figures 7 and 8 together, we conclude that **SAM is robust to different configurations of $(\eta,\rho)$ while USAM is robust to neither of them**. We will make our conclusion more transparent in the final version.
>
> ------
>
> ### More Practical Setting than "Adversarial Checkpoint"?
>
> Thanks for noticing this! We used the "Adversarial Checkpoint" by Damian et al. (2021) as a "poor global minimum" which has 100% training accuracy but only 48% test accuracy. This initialization **helps highlight different algorithm's behavior when they are close to "bad minimum"**. If we use a "good minimum" instead, we would not see the difference in behaviors as clear.
>
> For better understanding, we also conducted the same experiments but instead initialized from a "full-batch checkpoint" (Damian et al., 2021), which is the 100% training accuracy point reached by running full-batch GD on the training loss function. Such initialization is thus more realistic compared to the "adversarial checkpoint".
>
> The training trajectory initialized from this "full-batch checkpoint" is included in the attachment. In short, **USAM still gets stuck, while SAM attains better test accuracy via drifting along the minima manifold**. We will include this new experimental result in our final version.
>
> > ### References
> > * Alex Damian, Tengyu Ma, and Jason D. Lee. Label Noise SGD Provably Prefers Flat Global Minimizers. In *Advances in Neural Information Processing Systems*, 2021.
>
> ---
>
> ### Better Learning Rate Scheduling?
>
> In practice, there are various ways of tuning learning rates, e.g., linear or cosine scheduling. However, such tunings are a bit tricky to analyze in theory -- none of the previous papers has succeeded.
>
> Regarding the "no perturbation in the early stage" variant of USAM, we believe it probably avoids the divergence issue. However, this only reflects our results in Section 2. In Section 3 (or more specifically, Figure 5), we illustrate that **"USAM without perturbation in its early phase" cannot guarantee a good performance, either**.
>
> For more context, in Figure 5, we initialize all algorithms from the same "poor global minimum" derived by SGD and study which algorithms can escape it. Essentially, to arrive at this "poor global minimum", all algorithms are not adopting perturbation in their early stages. However, as suggested by our plot, USAM loses the ability to find better global minima, matching our conclusions in Theorems 5 and 6. This means that **normalization is not only important in the initial stages, but also critical in the final phase for better minima**. Consequently, removing perturbation in the early phase of USAM only resolves the divergence issue but still cannot ensure better performance.
>
> ---
>
> ### Does the 'USAM versus SGD' Result in [3] Contradit Ours?
>
> Thanks for mentioning [3]! In [3], the authors show that while USAM (which they call SAM) exhibits a similar EoS phenomenon as GD, **USAM has a smaller EoS threshold**: `We will show that SAM also induces an EOS stabilization effect, but at a smaller eigenvalue than GD` (from their Section 2.2.2). Hence, their result also precisely shows that USAM is less stable than GD: When the learning rate $\eta$ exceeds the EoS threshold of USAM but is below the one of GD (i.e., the condition $\eta \in (2/(\beta+\rho\beta^2),2/\beta]$ in our Theorem 1), USAM diverges, but GD does not. As a result, **their observation essentially supports our claim**.
>
> Meanwhile, we would like to mention that our early-stage analyses additionally reveal that SAM does not diverge for the range of $\eta$ where USAM diverges and is thus more robust to the choice of $\eta$, which is beyond the scope of [3] as [3] only studied USAM instead of SAM.
>
> ---
>
> Thank you once again for taking the time to review our paper. Given that you have not raised any concerns regarding the correctness or novelty of our work, if you believe that we have addressed your points, we would appreciate it if you would consider raising your score.

---

> > ### Comment · Reviewer_chBL · 2023-08-17
> >
> > Thank you for your response to the specific questions.
> >
> > I agree that the idea of comparing USAM and SAM is to emphasize the role of normalization, which is sometimes ignored. However, I also concur that the primary focus of USAM in previous studies has been to facilitate interpretation through the lens of SDE or mathematical optimization properties.
> >
> > As a reviewer, I am inclined to recommend a 'Weak Accept' for the paper. The central idea is lucid and well-anchored in mathematical clarity, despite some limitations in the experimental aspects.

---

### Official Review · Reviewer_pAM7 · 2023-07-06

**Soundness:** 4 excellent
**Presentation:** 3 good
**Contribution:** 1 poor
**Rating:** 7
**Confidence:** 4

**Summary:**

**I have read the rebuttal and raised the score to a 7.** This paper studies the importance of the normalization step in SAM (normalizing the gradient used to perturb the weights) for stability and effectiveness at finding a flat region in a manifold of minima (drifting along a continuum of minima). They provide the following theoretical explanations:
- On a strongly convex, smooth loss and some specific nonconvex settings, for the same range of learning rates, unnormalized SAM may diverge while normalized SAM does not i.e. the SAM perturbation has to be sufficiently small for convergence. Normalized SAM stabilizes within some local neighborhood around the minimizer (this has been shown previously in Bartlett 2022).
- Considering the model x*y for some convex loss, where the flattest solution lies at x,y=0, they show that USAM and GD both get stuck after reaching a closer sharper solution (x=0, y > 0), while SAM keeps drifting towards the origin after getting close to a minima (x=0, y > 0). Their contribution here is showing that without the normalization step, the perturbations become too small.

For these reasons, they empirically show (in a sparse coding problem) that the normalization step of SAM allows SAM to stably achieve flat minima in comparison to large learning rate GD or unnormalized SAM which may both have problems with divergence.

**Strengths:**

**Significance:** The unnormalized SAM has been studied theoretically as a simplification for SAM, and the contributions of the paper emphasize why this simplification may change the dynamics of SAM entirely.
**Clarity/quality:** The paper is well motivated, easy to read, and thorough. All theoretical conclusions are sound and are supported by clear empirical trends.

**Weaknesses:**

The main weakness is significance as unnormalized SAM which the paper focuses on studying is not utilized in practice and the conclusions of the work, though confirms normalization is good, does not provide further prescriptive insight. The characterizations of normalized SAM conducted in the paper have already been shown in previous works, and the new contribution is solely characterizing how unnormalized SAM is different. Yet, unnormalized SAM only appears in a couple works that theoretically analyze SAM but does not appear in practice. As pointed out by the authors, previous works (Wen 2022, Bartlett 2022) have already shown that normalized SAM stabilizes around a local neighborhood around a minima and it achieves flatter minima by continuing to drift along the manifold. The contribution of this work is to formalize how the perturbation needs to be sufficiently small for convergence, but sufficiently large around the manifold of minimas to find the flat solution, which is achieved by normalization. Yet these conclusions, especially the former, seems a bit intuitive/obvious. Additionally, previous theoretical works have indicated that USAM can be tuned to achieve similar performance to SAM (Andriuschenko 2022). Though there exists a difference between USAM and SAM in the toy models in the paper, there aren't any experiments that indicate that not normalizing is a problem on real datasets and models (that USAM requires delicate tuning, or doesn't work at all in terms of achieve similar training curve and test performance as SAM). Perhaps adding such an experiment would help amend the first concern.

--
Minor fixes:
Line 248 diverges --> divergence

**Questions:**

1. The theorem 4, the only place $\rho$ shows up is in the final phase, where $|x_t|, |y_t|$ are upper bounded by a factor proportional to $\rho$, as SAM stabilizes around a local neighborhood. Does this imply that SAM converges around the origin regardless of the magnitude of $\rho$ and $\rho$ only determines the rate of convergence?

2. In Figure 7, USAM with $\eta=0.1$ seems to do comparatively to SAM $\eta=0.1$. From this, it is a bit unclear whether it is truly difficult to tune the hyperparameters for USAM to work. Is it often the case that USAM can be tuned easily to achieve similar performance as SAM?
For example in  (Andriuschenko 2022), the training and test curves of SAM and USAM are relatively similar (e.g. Figure 22).

3. What is the dataset/model used in Figure 1?

---

> ### Author Rebuttal · Authors · 2023-08-09
>
> Thank you for your insightful comments! Here are our responses:
>
> ---
>
> ### USAM is Not Practical
>
> We agree that USAM is by no means practical. However, the reason for comparison is solely to understand SAM better and doesn't have to do with USAM's impracticality. In particular, we are trying to **understand the success of SAM in practice rigorously and theoretically**: Undoubtably, SAM exhibits ground-breaking success in practice (and likely thus amassing ~700 citations in ~2 years); thus, better understanding SAM is critical for designing better algorithms in the future.
>
> Indeed, extensive effort has been made study SAM in the theoretical ML literature. However, we notice that **most of such works assume that the normalization of SAM can be omitted "for simplicity"** (e.g., Andriushchenko and Flammarion (2022); Behdin and Mazumder (2023); Agarwala and Dauphin (2023); Kim et al. (2023)). However, as far as we know, **the only rationale behind such a simplification is just one experiment** on CIFAR10 conducted by Andriushchenko and Flammarion (2022).
>
> In our paper, we take a step back and **challenge this common "simplification", asking whether normalization plays a critical role in SAM**. To make our results scientifically rigorous, our paper considers the variant of SAM that **only** removes normalization, namely USAM. This provides a precise and scientific ablation study. Hence, **we did not consider USAM for its practicality** but **for providing a precise and scientific understanding of SAM**. By considering USAM, we successfully and rigorously identify (one of) the reasons behind SAM's success, namely the normalization step.
>
> Finally, we would like to remark that **previous papers do not yet achieve such an accurate identification**. For example, as you mentioned, Wen et al. (2023) show that SAM can drift along the minima manifold for a "flatter" minimum. However, the crucial role played by the normalization step is beyond the scope of their paper. In our work, by comparing USAM and SAM, we identify its crucial role.
>
> > ### References
> > * Maksym Andriushchenko and Nicolas Flammarion. Towards Understanding Sharpness-Aware Minimization. *Proceedings of the 39th International Conference on Machine Learning*, PMLR 162:639-668, 2022.
> > * Kayhan Behdin and Rahul Mazumder. On Statistical Properties of Sharpness-Aware Minimization: Provable Guarantees. *arXiv preprint arXiv:2302.11836*, 2023.
> > * Atish Agarwala, Yann Dauphin. SAM operates far from home: eigenvalue regularization as a dynamical phenomenon. *Proceedings of the 40th International Conference on Machine Learning*, PMLR 202:152-168, 2023.
> > * Hoki Kim, Jinseong Park, Yujin Choi, and Jaewook Lee. Stability Analysis of Sharpness-Aware Minimization. *arXiv preprint arXiv:2301.06308*, 2023.
> > * Kaiyue Wen, Tengyu Ma, and Zhiyuan Li. How Sharpness-Aware Minimization Minimizes Sharpness? In *The Eleventh International Conference on Learning Representations*, 2023.
>
> ---
>
> ### Non-Toy Experiments
>
> Thanks for mentioning this! We want to point out that **our experiments in Figure 5 are done on practical ResNet18 architecture on the CIFAR-10 dataset**. In Figure 5, we initialized SGD, SAM, and USAM at the same checkpoint, which is a poor global minimum of the loss function (which we call "adversarial checkpoint"; the same experiment is also conducted from a more realistic checkpoint yielded by full-batch GD, which is included in the attached PDF file).
>
> **From Figure 5, we observe that USAM is indeed harder to tune in real-world settings**: A conservative configuration of $(\eta,\rho)$ makes it unable to drift along the manifold (and is thus trapped at the bad minimum); an aggressive configuration drives it away from the manifold and results in worse performance. In sharp contrast, SAM keeps drifting along the manifold towards better-generalizing minima and achieves much better performance.
>
> ---
>
> ### Role of $\rho$ in Theorem 4
>
> Thanks for raising such a good question! In fact, $\rho$ also determines the convergence speed in the "Middle Phase". In Footnote 2, we mention that the dynamics of $y_t$ can be upper bounded by either $y_{t+1}\lesssim (1-\eta \rho^2)y_t$ or $y_{t+1}\lesssim y_t-\eta \rho$. Thus, a **larger $\rho$ may drive $y_t$ faster towards the origin**. We will include this as a remark after Theorem 4 to avoid possible confusion.
>
> And as a final remark, we would like to mention that **we made no effort to optimize the convergence rate bound**. The objective of Theorem 4 is to show that the trajectory will approach a neighborhood of the origin "pretty fast". It remains open how many iterations are needed for SAM to find such a neighborhood.
>
> ---
>
> ### USAM versus SAM in Figure 7 when $\eta=0.1$
>
> Indeed, when picking $\eta=0.1$ in Figure 7, the performance of USAM and SAM are similar. Meanwhile, for other choices of $\eta$ (e.g., $0.025$), USAM performs much worse while SAM remains good.
>
> However, this tendency is not valid in general. As presented in Figure 8 of Appendix B, when setting the algorithmic parameters $(\eta,\rho)$ too aggressively, USAM diverges (the $\rho=0.5$ curve becomes invisible in the plot because it diverges). Thus, **tuning the parameters of $\eta$ and $\rho$ is indeed a dilemma** in USAM, just as in GD.
>
> ---
>
> ### Dataset in Figure 1
>
> In Figure 1, we adopt the over-parametrized matrix sensing problem setup introduced by Li et al. (2018). The exact setup can be found in Appendix A.
>
> > ### References
> > * Yuanzhi Li, Tengyu Ma, and Hongyang Zhang. Algorithmic regularization in over-parameterized matrix sensing and neural networks with quadratic activations. In *Conference On Learning Theory*, pages 2–47. PMLR, 2018.
>
> ---
>
> We thank you once again for such a valuable review of our paper. We are more than happy to answer any further questions!

---

### Official Review · Reviewer_ymqN · 2023-07-07

**Soundness:** 4 excellent
**Presentation:** 3 good
**Contribution:** 3 good
**Rating:** 7
**Confidence:** 3

**Summary:**

The authors argue that the normalization of the gradient in the SAM model during the ascent step is beneficial. In particular, classic SAM is shown theoretically and empirically in certain scenarios to not-diverge and to be able to drift along the minima manifold. This implies that SAM is more robust in the choice of the step-sizes compared to the unnormalized SAM.

**Strengths:**

- The paper is in general well-written and accessible despite being a theoretical work.
- The included examples help the understanding of the ideas.
- The theoretical results seem convincing and sensible, but I have not verified the technical details.

**Weaknesses:**

- I think the similarities and differences to Wen 2023 and Compagnoni 2023 could have been discussed more broadly.

**Questions:**

Q1. For the non-convex and L-smooth case (Sec 2.2) SAM does not diverge for any function with the same properties? Or is it possible for SAM to diverge in some functions of this class?

Q2. Regarding the function $\mathcal{L}$ in Sec 3.1. Does the $\times$ imply multiplication of $x,y$? The function $\mathcal{L}$ is $L$-smooth?

Q3. Theorem 6 is compared with Wen 2023 where the step-sizes are ``small enough''. Does this result generalize in the setting of the current paper where the argument is that discrete step-size selection is more robust?

**Limitations:**

The authors do not discuss any potential limitation of the approach. Implicitly, the limitations are encoded in the assumptions of the theorems. Also, the paper proposes a technical method with no direct negative societal impact.

---

> ### Author Rebuttal · Authors · 2023-08-09
>
> Thank you for your insightful comments! Here are our responses:
>
> ---
>
> ### Comparison Between Our Paper and Wen et al. (2023); Compagnoni et al. (2023)
>
> Thanks for mentioning these two seminal papers on SAM! We will definitely discuss more in our revision. Here is a draft of our discussion:
>
> 1. Wen et al. (2023) characterized the behavior of SAM when *1)* initialized near the minima manifold, and *2)* the step size $\eta$ is small enough, by showing that SAM roughly follows a Riemannian gradient flow on $\lambda_{\max}(\nabla^2 \mathcal L(w))$ (i.e., their "sharpness" measure maximum eigenvalue of Hessian) along the minima manifold. **This result is closely related to ours in Section 3** as we also study the behavior of SAM when close to a global minimum.
> However, there are also several differences:
>     * Admittedly, they consider more general loss functions than ours. However, as a side-effect, they require "small enough" $\eta$ which they do not specify. They also require the initialization to be sufficiently close to the minima manifold.
>     * In contrast, we allow any $\eta=o(1)$ and $\rho=\mathcal O(1)$ (we include detail requirements in Theorem 16) and focus on a specific loss function. **This gives a much more accurate characterization of the algorithmic behavior of SAM**, even when the initialization is far from the origin.
>
> 2. Compagnoni et al. (2023) consider the continuous-time behavior of SGD, SAM, and USAM. As we mentioned in the related work section, this is the only work (to our knowledge) that distinguishes USAM from SAM. **Indeed, they also show different behaviors of SAM and USAM** (i.e., USAM is attracted to local minima while SAM aims at global ones). However, as they are considering continuous-time variants of algorithms while we consider discrete (original) versions, **our results directly apply to the SAM in practice and the USAM in theory**.
>
> ---
>
> ### Does SAM Diverge in Section 2.2 (Non-Convex but Smooth Loss Functions)?
>
> Yes, it is possible for SAM to diverge when facing non-convex loss functions, similar to GD which diverges when $\eta$ is too large.
>
> However, we would like to emphasize that the purpose of Section 2 is to illustrate that in the early stage of training, *1)* SAM is **more robust than USAM** as USAM diverges even with a rather small $\eta$ or $\rho$, and *2)* SAM has **comparable robustness to GD** as an $\eta$ ensuring convergence of GD can also make SAM converge.
>
> In other words, one of our main messages in that section is **"SAM is better than USAM", not "SAM is better than GD"**. The superiority of SAM over GD is left for Section 3, where SAM exhibits the ability to drift along the minima manifold (which is beneficial as suggested by various previous works, both theoretically and empirically) while GD easily gets trapped at "bad" minima.
>
> ---
>
> ### Function $\mathcal L$ in Section 3.1
>
> Yes, $\times$ means the multiplication. When restricting $\mathcal L$ in a compact region, it becomes $L$-smooth.
>
> ---
>
> ### "Small Enough" Step Sizes in (Wen et al., 2023)
>
> Thanks for referring to this subtle difference. Indeed, the conclusion by Wen et al. (2023) requires a "small enough" step size for the dynamics of SAM to track a Riemannian gradient flow induced by $\lambda_{\max}(\nabla^2 \mathcal L(w))$ and thus decrease the "sharpness".
>
> Nevertheless, from our Theorem 4 (behavior of SAM on single-neuron networks) and Figure 5 (behavior of SAM on ResNet18 over CIFAR-10), **we conjecture that when the step sizes are not "small enough", SAM can still drift along the manifold** and thus contrasts USAM's behavior characterized in Theorem 6. Proving or disproving this would be an interesting future direction.
>
> ---
>
> Thank you once again for taking the time to review our paper. Given that you have not raised any concerns regarding the correctness or novelty of our work, if you believe that we have addressed your points, we would appreciate it if you would consider raising your score.

---

> > ### Comment · Reviewer_ymqN · 2023-08-21
> > **Post-rebuttal**
> >
> > I would like to thank the authors for their replies. In light of the other reviews, I will keep my score and vote for acceptance.

---

### Author Rebuttal · Authors · 2023-08-09

Dear Reviewers,

We thank all the reviewers for the valuable feedback. Specifically, thank you for appreciating that our motivation is "**clear and interesting**" (Reviewers  pAM7, chBL, and a57m), our paper is "**well-written, easy-to-read, and thorough**" (Reviewers ymqN, pAM7, and a57m), the theoretical analysis is "**convincing and sound**" (Reviewers ymqN, pAM7, chBL, and a57m), the empirical results "**clearly support our key takeaways**" (Reviewers ymqN, pAM7, FoGd, and a57m), our paper "**identifies the role of normalization and highlights its importance**" (Reviewers pAM7, chBL, and FoGd).

Here, we first dicuss the two main concerns shared by the reviewers, namely *i) why consider USAM (which seems to be impractical)*, and *ii) what are the main takeaways of our paper*. More detailed responses are included in our responses to Reviewer a57m and Reviewer FoGd, respectively.

1. *Why consider USAM?*

We agree that USAM is not practical, but **we consider it in order to understand SAM better**. To provide brief context, despite the success of SAM, its theory is at its nascent stage, and many existing theoretical works _ignore the normalization step_ in SAM "for simplicity."
Our work is motivated precisely by this state of affairs in the theory of SAM, and our work reveals that such "for simplicity" choice is unwarranted,  as **normalization plays an important role** in SAM.

Moreover, in order to conduct a principled ablation based investigation, **we only remove normalization in SAM, and keep all other parts unchanged**. This reason underlies **the reason for considering USAM;** we are not claiming USAM's practicality (or any other advantage).

2. *What are the main takeaways of our paper?*

There are **two main takeaways,** one for the theoretical ML community, and another one for the empirical ML community.

For the theoretical community, our takeaway is that **normalization plays a critical role and cannot be neglected when analyzing SAM**. We believe this message should be valuable to the community, as various previous theoretical papers have omitted normalization -- we demonstrate that such an omission drastically changes the algorithmic behavior of SAM.

For the empirical community, we find **the success of SAM is (at least in part) backed up by the normalization** -- although SAM exhibits  great success in practice, and the success is supported by various theoretical works, its working mechanism is far from understood. We start this investigation by specifically focusing on the importance of normalization.

---

We also thank the reviewers for catching the confusing arguments in our draft -- we will update them in the final version as promised. Many thanks to all the reviewers for your efforts.

Finally, we also included an additional plot as an attachment. Its explanation can be found in our second response to Reviewer chBL.

We hope our response resolves your main concerns. If you have any further questions, please do not hesitate to contact us. We are more than happy to answer them!

Thank you and Best Regards,
Authors

---

### Decision · Program_Chairs · 2023-09-21

**Decision:**

Accept (poster)

**Comment:**

The authors investigate the importance of normalization in the Sharpness-Aware Minimization (SAM) algorithm. They do this by comparing how gradient descent, SAM, and unnormalized SAM behave through experiments and theory. The main finding is that normalization is crucial and must not be overlooked when analyzing SAM. Reviewers consider the work relevant, with valuable theoretical outcomes. They've also provided suggestions to enhance the paper's presentation. We recommend authors to include these in the final version. In summary, I recommend accepting this paper.